# REM sleep promotes experience-dependent dendritic spine elimination in the mouse cortex

Yanmei Zhou[1,2,3,9], Cora Sau Wan Lai [4,5,9], Yang Bai[3], Wei Li[6], Ruohe Zhao[1,2], Guang Yang [7], Marcos G. Frank[8] & Wen-Biao Gan [2✉]

In many parts of the nervous system, experience-dependent refinement of neuronal circuits predominantly involves synapse elimination. The role of sleep in this process remains unknown. We investigated the role of sleep in experience-dependent dendritic spine elimination of layer 5 pyramidal neurons in the visual (V1) and frontal association cortex (FrA) of 1-month-old mice. We found that monocular deprivation (MD) or auditory-cued fear conditioning (FC) caused rapid spine elimination in V1 or FrA, respectively. MD- or FC-induced spine elimination was significantly reduced after total sleep or REM sleep deprivation. Total sleep or REM sleep deprivation also prevented MD- and FC-induced reduction of neuronal activity in response to visual or conditioned auditory stimuli. Furthermore, dendritic calcium spikes increased substantially during REM sleep, and the blockade of these calcium spikes prevented MD- and FC-induced spine elimination. These findings reveal an important role of REM sleep in experience-dependent synapse elimination and neuronal activity reduction.

[1] School of Chemical Biology and Biotechnology, Peking University Shenzhen Graduate School, Shenzhen 518055, China. [2] Skirball Institute, Department of Neuroscience and Physiology, Department of Anesthesiology, New York University School of Medicine, New York, NY 10016, USA. [3] Shenzhen Bay Laboratory, Shenzhen 518132, China. [4] School of Biomedical Sciences, Li Ka Shing Faculty of Medicine, The University of Hong Kong, Pokfulam, Hong Kong Special Administrative Region, China. [5] State Key Laboratory of Brain and Cognitive Sciences, The University of Hong Kong, Pokfulam, Hong Kong Special Administrative Region, China. [6] School of Pharmaceutical Sciences (Shenzhen), Sun Yat-sen University, Guangzhou 510275, China. [7] Department of Anesthesiology, Columbia University, New York, NY 10032, USA. [8] College of Medical Sciences, Washington State University, Spokane, WA 99201, USA. [9] These authors contributed equally: Yanmei Zhou, Cora Sau Wan Lai. ✉email: gan@saturn.med.nyu.edu

Sleep is an important physiological process that occupies at least one third of human life. Mammalian sleep cycles compose of non-rapid eye movement (NREM) sleep and rapid eye movement (REM) sleep[1,2]. Many lines of evidence suggest that sleep is important for regulating neuronal plasticity during brain development and after learning[3–9]. For example, NREM sleep has been shown to promote synaptic potentiation[10], synapse formation[11], and synaptic weakening after wakefulness[12–14]. REM sleep has been found to increase the expression of synaptic plasticity-related genes[15–18], and selectively maintain newly formed dendritic spines induced after motor learning[19]. In addition, REM sleep deprivation reduces neuronal excitability[20,21], the induction and maintenance of long-term potentiation[22–24], as well as ocular dominance plasticity in the developing cat visual cortex[15]. Together, these studies support a view that both NREM and REM sleep have important roles in activity- and experience-dependent synaptic plasticity.

A prominent feature of experience-dependent synaptic plasticity is the pruning of existing synapses during development and adulthood[25–29]. Indeed, recent studies have shown that monocular deprivation (MD) or auditory-cued fear conditioning (FC) significantly increases dendritic spine elimination of layer 5 pyramidal neurons in the developing mouse primary visual cortex (V1) or frontal association cortex (FrA), respectively[29,30]. Such experience-dependent dendritic spine elimination likely contributes to changes of neuronal activity in response to visual and auditory inputs[27,31,32]. While studies so far have shown that sleep is important for learning-dependent dendritic spine formation

and maintenance[11,19], as well as dendritic spine remodeling in the developing sensory cortex[33,34], it remains unknown whether sleep is involved in experience-dependent synapse elimination.

In this study, we investigated the role of sleep in dendritic spine elimination induced by MD and FC in the developing V1 and FrA, respectively. We found that in both cortical regions, total sleep or REM sleep deprivation decreased experience-dependent elimination of dendritic spines and reduction of neuronal activity of layer 5 pyramidal cells. We also found that dendritic $Ca^{2+}$ spikes increased during REM sleep and were involved in dendritic spine elimination. These findings suggest the important role of REM sleep in experience-dependent synapse elimination, likely via dendritic $Ca^{2+}$ spike-dependent mechanisms.

## Results

**SD or REMD reduces MD-induced spine elimination in V1.** A previous study has shown that 3-day MD results in dendritic spine elimination on apical dendrites of layer 5 pyramidal neurons in the binocular region of the developing mouse primary visual cortex (V1)[30]. To investigate whether sleep plays a role in this process, we first determined the time course of MD-induced dendritic spine remodeling of layer 5 pyramidal neurons in the V1 of 4-week-old mice. Transcranial two-photon microscopy was used to repeatedly image dendritic spines over 4–24 h in awake, head-restrained transgenic mice expressing yellow fluorescent protein (YFP) in layer 5 pyramidal neurons (Fig. 1a). We found that MD significantly increased dendritic spine elimination within

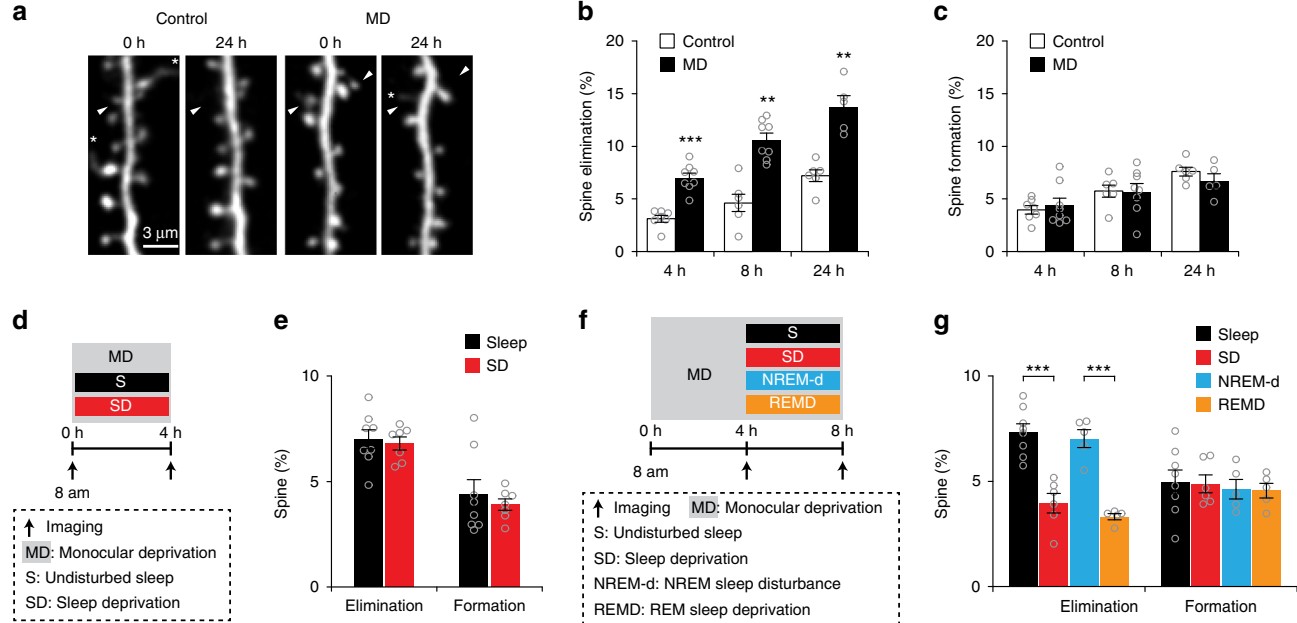

**Fig. 1 SD or REMD reduces MD-induced spine elimination in V1. a** Repeated imaging of dendritic spines on apical tuft dendrites of layer 5 pyramidal neurons in MD and non-deprived control mice. White arrowheads indicate eliminated spines, asterisks indicate filopodia. Scale bar: 3 μm. **b** Dendritic spine elimination was significantly higher in MD mice than in control mice over 4–24 h (control: $n = 7$, 6 and 6 mice for 4, 8, and 24 h, respectively; MD: $n = 8$, 8 and 5 mice for 4, 8, and 24 h, respectively; $P = 0.0003$, $= 0.0013$ and $= 0.0043$ for 4, 8, and 24 h, respectively, Wilcoxon–Mann–Whitney test, two-sided). **c** No significant difference in dendritic spine formation between MD and control mice (control: $n = 7$, 6 and 6 mice for 4, 8 h and 24 h, respectively; MD: $n = 8$, 8 and 5 mice for 4, 8, and 24 h, respectively; $P = 0.817$, $= 1.000$, and $= 0.361$ for 4, 8, and 24 h, respectively, Wilcoxon–Mann–Whitney test, two-sided). **d** Schematic of experimental design to assess the effect of sleep on MD-induced spine remodeling. **e** No significant difference in spine elimination ($P = 0.487$) or formation ($P = 0.908$) was observed in mice with or without sleep during the 0–4 h MD period (Sleep: $n = 8$ mice, SD: $n = 7$ mice, Wilcoxon–Mann–Whitney test, two-sided). **f** Schematic of experimental design. After the initial 4-h MD, mice were subjected to an additional 4-h MD with or without sleep/REM sleep to assess the effect of sleep. **g** SD or REMD during the second 4 h significantly reduced MD-induced dendritic spine elimination but not spine formation as compared to that in sleep or NREM-d mice, respectively (SD: $n = 6$ mice, Sleep: $n = 8$ mice, REMD: $n = 5$ mice, NREMD-d: $n = 5$ mice, $P = 0.0005$, Kruskal–Wallis test followed by multiple comparisons test). **P < 0.01, ***P < 0.001. Data are presented as mean ± s.e.m. Source data for (**b**), (**c**), (**e**), and (**g**) are provided as a Source Data file.

4–24 h in the binocular region of V1 as compared to non-deprived control (Fig. 1a, b; see also Supplementary Movies 1–4, Control: 1356 spines from 7 animals (4 h); 1040 spines from 6 animals (8 h); 1166 spines from 6 animals (24 h); MD: 1409 spines from 8 animals (4 h); 1376 spines from 8 animals (8 h); 828 spines from 5 animals (24 h)). In contrast, the rate of spine formation was not significantly altered after MD over 4–24 h (Fig. 1a, c). These results indicate that MD leads to rapid elimination of dendritic spines of layer 5 pyramidal neurons during the critical period of visual cortex development.

To examine the potential involvement of sleep in MD-induced dendritic spine elimination, we compared dendritic spine remodeling in mice with undisturbed sleep and with sleep deprivation (SD). SD was carried out by gentle handling between two imaging sessions (Fig. 1d). Electroencephalography (EEG) and electromyography (EMG) recordings over 4 h showed that SD mice were awake $94.5 \pm 1.4\%$ of the time, whereas mice with undisturbed sleep were awake $44.1 \pm 3.3\%$ of the time (Supplementary Figs. 1 and 2a). We found no significant difference in MD-induced spine elimination over 4 h between mice with undisturbed sleep or SD (Fig. 1e; Sleep: 1409 spines from 8 animals; SD: 1319 spines from 7 animals). The rate of spine formation was also comparable between the two groups (Fig. 1e). These results suggest that sleep has no significant effect on spine elimination or formation during the first 4 h of MD.

Notably, after the initial 4-h MD, when mice were subjected to an additional 4-h MD and deprived of total sleep only during the second 4-h period (Fig. 1f and Supplementary Fig. 2b), we found that the rate of spine elimination was significantly lower in SD mice than mice with undisturbed sleep (Fig. 1g; Sleep: 1376 spines from 8 animals; SD: 1122 spines from 6 animals). In contrast, the rate of spine formation was comparable between mice with or without SD (Fig. 1g). These results indicate that after an initial 4-h MD, subsequent sleep promotes MD-induced spine elimination in the developing V1.

Sleep consists of NREM and REM sleep stages[1,2]. To examine whether REM sleep is involved in MD-induced spine elimination, mice were subjected to MD and deprived of REM sleep (REMD) for 4 h after the initial 4-h MD (Fig. 1f). REM sleep was monitored continuously by EEG and EMG recordings and disrupted by gentle handling upon detection (Supplementary Fig. 1). To control for non-specific effects related to gentle handling, a group of mice were disturbed during the NREM sleep period the same number of times as that for REMD (named as NREM sleep disturbed (NREM-d) mice) (Fig. 1f). EEG/EMG recordings over 4 h showed that REM sleep was significantly reduced in REMD mice ($0.9 \pm 0.2\%$) when compared to mice with undisturbed sleep ($6.2 \pm 0.7\%$) and NREM-d mice ($5.0 \pm 0.5\%$) (Supplementary Fig. 2b, c). We found that MD-induced spine elimination was significantly lower in REMD mice than in NREM-d mice, while the rate of spine formation was comparable between the two groups (Fig. 1g; REMD: 851 spines from 5 animals; NREM-d: 876 spines from 5 animals). Furthermore, in mice without MD, REMD during the second 4-h period had no effect on either dendritic spine elimination or formation when compared to mice with undisturbed sleep (Supplementary Fig. 3; Sleep: 859 spines from 5 animals; REMD: 982 spines from 5 animals). Together, these results suggest that after an initial 4-h MD, subsequent REM sleep promotes MD-induced dendritic spine elimination of layer 5 pyramidal neurons in the developing V1.

**SD or REMD prevents MD-induced neuronal activity reduction.** A previous electrophysiological study has reported that sleep enhances the effect of MD on neuronal responses in the developing cat visual cortex[35]. To investigate whether sleep or REM sleep affects MD-induced changes of neuronal activity in the developing mouse visual cortex, we performed $Ca^{2+}$ imaging of layer 5 pyramidal neurons expressing genetically encoded $Ca^{2+}$ indicator GCaMP6s. In this experiment, mice were head-restrained and the previously deprived eye was exposed to visual stimulus under a two-photon microscope. The somatic activity of layer 5 pyramidal neurons in the binocular region of the visual cortex contralateral to the deprived eye was measured during the visual stimulation over a period of 60 s (Fig. 2a). In mice subjected to an earlier 4-h MD, neuronal activity evoked by visual stimulation of the deprived eye was significantly decreased after an additional 4-h MD and undisturbed sleep during the second 4-h period (Fig. 2a–c and Supplementary Fig. 4). Notably, SD or REMD in the second 4-h period prevented the reduction of neuronal response in the V1 corresponding to the deprived eye (Fig. 2a, d–g and Supplementary Fig. 4). Thus, REM sleep not only promotes MD-induced dendritic spine elimination but also reduces the activity of layer 5 pyramidal neurons in the binocular region of V1 contralateral to the deprived eye.

**SD or REMD reduces FC-induced spine elimination in FrA.** It has been shown that auditory-cued FC induces dendritic spine elimination of layer 5 pyramidal neurons over days in the mouse frontal association (FrA) and motor cortices[29,31]. To investigate whether sleep is also involved in spine elimination induced by FC, we first determined the time course of FC-induced spine remodeling of layer 5 pyramidal neurons in the FrA of 1-month-old mice. We observed that both elimination and formation of dendritic spines were significantly higher 4–8 h after FC in mice receiving auditory cues paired with foot shocks as compared to control mice receiving unpaired stimuli (Fig. 3a–c; Unpaired control: 1385 spines from 9 animals (4 h); 773 spines from 5 animals (8 h); 1382 spines from 9 animals (24 h); Paired: 1369 spines from 9 animals (4 h); 745 spines from 5 animals (8 h); 1337 spines from 9 animals (24 h)). Over 24 h, only the rate of spine elimination showed a significant increase after FC (Fig. 3a–c). As expected, 24 h after FC, mice subjected to FC exhibited a significantly higher level of conditioned freezing responses than control mice in a recall test (9 animals per group, Fig. 3d). Together, these data indicate that FC induces rapid dendritic spine remodeling within hours in FrA.

To investigate the potential role of sleep in FC-induced spine remodeling, mice were sleep deprived for 4 h by gentle handling immediately after FC (SD group), and dendritic spine remodeling were compared between mice with or without SD (Fig. 3e). We found that FC-induced spine elimination over 4 h was significantly lower in SD mice than in mice with sleep (Fig. 3f; Sleep: 758 spines from 5 animals; SD: 844 spines from 5 animals). We further compared FC-induced spine remodeling between mice deprived of REM sleep for 4 h and mice perturbed during NREM sleep period the same number of times as for REMD (NREM-d mice as in Fig. 1f). EEG and EMG recordings over 4 h showed that REM sleep was significantly reduced in REMD mice as compared to NREM-d mice (Supplementary Fig. 2d). Similar to SD mice, FC-induced spine elimination over 4 h was significantly lower in REMD mice than in NREM-d mice (Fig. 3f; REMD: 1000 spines from 5 animals; NREM-d: 738 spines from 4 animals). Furthermore, over 24 h, there was a significant reduction in spine elimination between mice with SD or REMD and mice with sleep or NREM-d (Supplementary Fig. 5). Consistent with the role of REM sleep in emotional memory processing[36–43], a recall test 24 h after FC showed a reduced freezing response in mice subjected to SD or REMD (Fig. 3g). Taken together, these results indicate that REM sleep is crucial for FC-induced dendritic spine elimination and freezing responses.

In addition to spine elimination, we found that FC-induced spine formation over 4 h was significantly lower in SD mice than in mice with sleep (Fig. 3f). However, there was no significant difference in

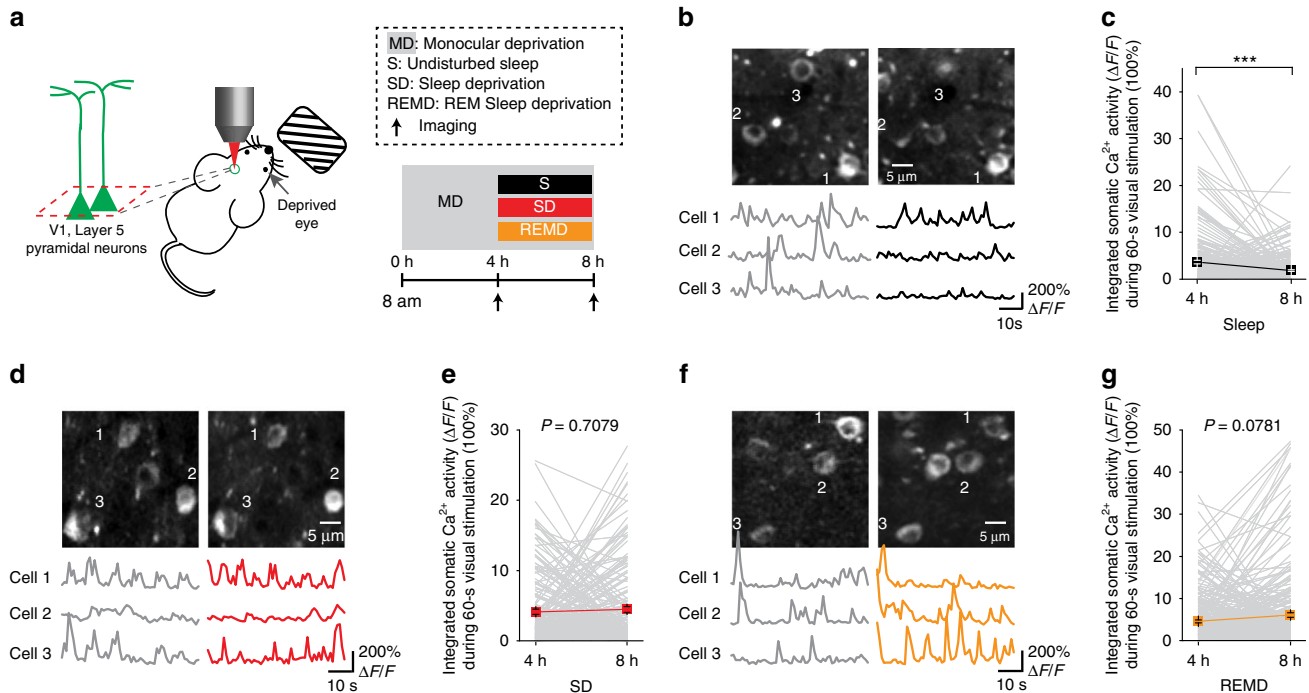

**Fig. 2 SD or REMD prevents MD-induced neuronal activity reduction. a** Schematic of experimental design. After the first 4-h MD, mice were subjected to an additional 4-h MD and were either allowed to sleep or subjected to sleep/REM sleep deprivation. Two-photon $Ca^{2+}$ imaging was performed to assess the effect of sleep/REM sleep on somatic activity of layer 5 pyramidal neurons in V1. **b** Somatic $Ca^{2+}$ activity of layer 5 pyramidal neurons in the V1 contralateral to the deprived eye before (left) and after (right) 4-h sleep. $\Delta F/F_0$ was summed to measure somatic $Ca^{2+}$ activity over 1 min. Scale bar: 5 μm. 3 traces: somatic $Ca^{2+}$ activities of cells on the top. $Ca^{2+}$ fluorescence traces over 60 s are shown. **c** The integrated somatic $Ca^{2+}$ activity during 60-s visual stimulation was significantly decreased after MD with 4-h undisturbed sleep ($n = 250$ cells from 4 mice, $P = 0.0002$, Wilcoxon signed-rank test, two-sided). **d** Somatic $Ca^{2+}$ activity of layer 5 pyramidal neurons in the V1 contralateral to the deprived eye before (left) and after (right) 4-h SD. **e** 4-h SD significantly prevented the reduction of neuronal activity ($n = 194$ cells from 4 mice, $P = 0.7079$, Wilcoxon signed-rank test, two-sided). **f** Somatic activity of layer 5 pyramidal neurons in the V1 contralateral to the deprived eye before (left) and after (right) 4-h REMD. **g** REMD for 4 h significantly prevented the reduction of neuronal activity ($n = 294$ cells from 3 mice, $P = 0.0781$, Wilcoxon signed-rank test, two-sided). ***$P < 0.001$. Data are presented as mean ± s.e.m. Source data for (**c**), (**e**), and (**g**) are provided as a Source Data file.

spine formation over 4 h between REMD and NREM-d mice (Fig. 3f). Over 24 h, no significant difference in spine formation was found among mice with sleep, SD, REMD or NREM-d (Supplementary Fig. 5). Thus, FC-induced spine formation is lower over 4 h, but not 24 h, in SD mice than REMD or NREM-d mice, suggesting a role of NREM sleep in transient spine formation after FC.

**SD or REMD prevents FC-induced neuronal activity reduction.** To test whether sleep or REM sleep affects not only FC-induced spine elimination but also changes of neuronal activity, we performed $Ca^{2+}$ imaging of GCaMP6s-expressing layer 5 pyramidal neurons in FrA. In this experiment, awake head-restrained mice were subjected to the presentation of a conditioned auditory stimulus (CS) under a two-photon microscope, and somatic $Ca^{2+}$ activity of layer 5 pyramidal neurons was imaged during CS presentation over a period of 30 s before and 4 h after FC (Fig. 4a). In mice with 4 h undisturbed sleep, we found that neuronal activity in response to CS presentation was significantly reduced after FC (Fig. 4a–c and supplementary Fig. 6; 309 cells from 4 mice). In contrast, somatic $Ca^{2+}$ activity of layer 5 pyramidal neurons did not exhibit a significant reduction after FC in either the SD group (Fig. 4a, d, e, and Supplementary Fig. 6; 122 cells from 5 mice) or REMD group (Fig. 4a, f, g, and supplementary Fig. 6; 195 cells from 3 mice). Thus, REM sleep after FC promotes not only the elimination of dendritic spines but also the reduction of neuronal activity of layer 5 pyramidal neurons in response to conditioned auditory cue in FrA.

**Dendritic $Ca^{2+}$ spikes increase during REM sleep.** A recent study in the mouse primary motor cortex suggests that dendritic $Ca^{2+}$ spikes increase during REM sleep and are important for selective pruning and strengthening of newly formed dendritic spines[19]. To investigate whether dendritic $Ca^{2+}$ spikes may mediate the effect of REM sleep on dendritic spine elimination, we performed $Ca^{2+}$ imaging to examine the activity of apical tuft dendrites of layer 5 pyramidal neurons expressing the genetically encoded $Ca^{2+}$ indicator GCaMP6s in V1 and FrA of head-restrained mice in various brain states under a two-photon microscope (Fig. 5a and Supplementary Fig. 7a). In both cortical regions, we observed a robust increase of dendritic $Ca^{2+}$ transients occurring across long stretches (> 30 μm) of apical tuft dendrites during REM sleep as compared to NREM sleep or quiet awake state (Fig. 5b, c; see also Supplementary Movies. 5, 6). On average, the frequency and integrated activity of dendritic $Ca^{2+}$ transients were significantly higher during REM sleep than during other brain states in both V1 and FrA (Fig. 5d, e, g and h). The duration of dendritic $Ca^{2+}$ transients was also significantly longer during REM sleep than those during other brain states in both cortical regions (Fig. 5f, i). Furthermore, these dendritic $Ca^{2+}$ transients during REM sleep resemble NMDAR-activation-dependent dendritic $Ca^{2+}$ spikes described previously in the motor cortex[19,44] and could be blocked by the NMDA receptor antagonist MK801 (Fig. 6a–e). Thus, similar to the motor cortex, V1 and FrA also show a large number of dendritic $Ca^{2+}$ spikes on apical dendrites of layer 5 pyramidal neurons during REM sleep.

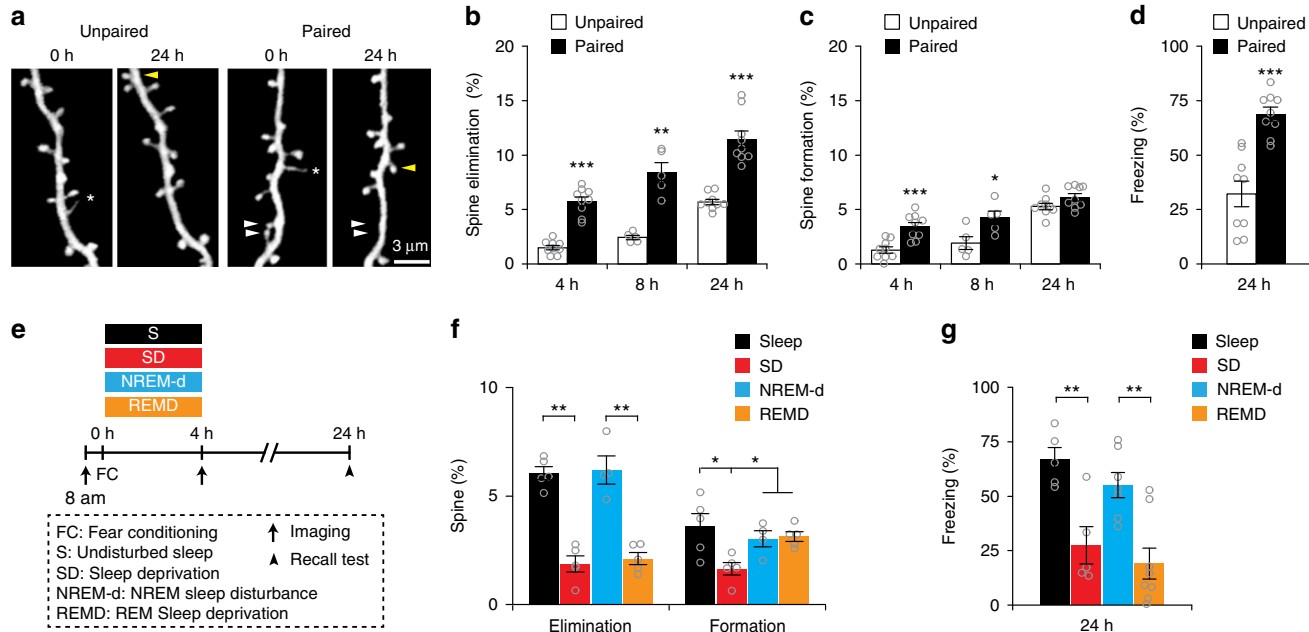

**Fig. 3 SD or REMD reduces FC-induced spine elimination in FrA. a** Repeated imaging of dendritic spines on the apical tuft dendrites of layer 5 pyramidal neurons in FrA in paired and unpaired mice. White and yellow arrowheads indicate eliminated and formed spines respectively; asterisks indicate filopodia. Scale bar: 3 μm. **b** Dendritic spine elimination was significantly higher in the paired group than unpaired group over 4–24 h ($n = 9$ mice at 4 h, 24 h for each group and $n = 5$ mice at 8 h for each group; $P < 0.0001$, $= 0.0079$ and $< 0.0001$ for 4, 8, and 24 h, respectively, Wilcoxon–Mann–Whitney test, two-sided). **c** Dendritic spine formation was significantly higher in the paired group than unpaired group within 8 h, but not over 24 h ($n = 9$ mice at 4 h, 24 h for each group and $n = 5$ mice at 8 h for each group; $P = 0.0006$, $= 0.0317$ and $= 0.0814$ for 4, 8, and 24 h, respectively, Wilcoxon–Mann–Whitney test, two-sided). **d** Freezing response test 24 h after FC ($n = 9$ mice for each group; $P < 0.0001$, Wilcoxon–Mann–Whitney test, two-sided). **e** Schematic of experimental design to assess the effect of sleep or REM sleep. **f** FC-induced dendritic spine elimination was significantly reduced after 4-h SD or REMD, while spine formation was significantly reduced only after 4-h SD (Sleep, SD and REMD: $n = 5$ mice for each group, NREM-d: $n = 4$ mice, $P = 0.0036$ for elimination and $P = 0.021$ for formation, Kruskal–Wallis test followed by multiple comparisons test). **g** SD or REMD significantly reduced freezing response 24 h after FC (SD and Sleep: $n = 5$ mice for each group; REMD: $n = 8$ mice, NREM-d: $n = 7$ mice, $P = 0.0024$, Kruskal–Wallis test followed by multiple comparisons test). $*P < 0.05$, $**P < 0.01$, $***P < 0.001$. Data are presented as mean ± s.e.m. Source data for (**b–d**), (**f**) and (**g**) are provided as a Source Data file.

**Blockade of dendritic Ca²⁺ spikes reduces spine elimination.**
Many lines of evidence indicate that dendritic $Ca^{2+}$ spikes are important for synaptic potentiation and depotentiation[45–49]. To examine whether dendritic $Ca^{2+}$ spikes during REM sleep are important for dendritic spine elimination induced by MD and FC, we puffed MK801 briefly and locally into the layer 1 of V1 or FrA at the beginning of REM sleep (Fig. 6a and Supplementary Figs. 7b and 8). Consistent with a recent study in the motor cortex[19], the number and integrated activity of dendritic $Ca^{2+}$ spikes occurring in REM sleep were substantially reduced when MK801 was locally injected in V1 or FrA at the beginning of REM sleep, but not during NREM sleep (Fig. 6b–e). Similar to REMD, MK801 blockade of dendritic $Ca^{2+}$ spikes during REM sleep reduced dendritic spine elimination over 4 h induced by MD or FC (Figs. 1g, 3f, 6f–i; MD: 702 spines from 4 animals; FC: 939 spines from 5 animals). In contrast, MK801 injection in V1 or FrA during NREM sleep had no significant effect on spine elimination induced by MD or FC (Figs. 1g, 3f, 6f–i; MD: 826 spines from 5 animals; FC: 1254 spines from 7 animals). There was no significant difference in spine formation between mice with MK801 injected during REM sleep and NREM sleep (Fig. 6f–i). Taken together, these results suggest that dendritic $Ca^{2+}$ spikes arising during REM sleep are involved in facilitating experience-dependent dendritic spine elimination.

## Discussion

Experience-dependent synapse elimination is an important mode of neuronal circuit refinement during development and after learning[25–29,50,51]. Our results show that REM sleep is critical for MD- and FC-induced elimination of dendritic spines in the developing visual and FrA, respectively. REM sleep is also important for the reduction of somatic activity of layer 5 pyramidal neurons in response to visual stimuli or conditioned auditory cues, as well as for the increased freezing response after FC. In addition, dendritic $Ca^{2+}$ spikes on apical tuft dendrites of layer 5 pyramidal neurons increase during REM sleep and are involved in experience-induced synapse elimination. These results reveal the important role of REM sleep in sensory and learning experience-dependent synapse elimination and reduction of neuronal activity.

Sleep has been associated with increased elimination of dendritic spines in the developing mouse sensory cortices[33,34] and decreased synaptic markers in Drosophila brain[52]. REM sleep is not required for new spine formation after motor training[11], but selectively strengthens some of these new spines to facilitate their long-term maintenance[19]. However, the role of sleep in experience-dependent elimination of existing synapses has not been examined previously. Our findings indicate that sensory deprivation- and FC-induced dendritic spine elimination in V1 and FrA depends on sleep. The effects of SD on MD- or FC-induced spine elimination of layer 5 pyramidal neurons are comparable to those of REMD, suggesting that REM sleep is critical for promoting experience-dependent synapse elimination. A recent study has shown that NREM sleep, not REM sleep, after motor learning promotes dendritic spine formation of layer 5 pyramidal neurons in the motor cortex[11]. In FrA, we found that dendritic spine formation over 4 h after FC is

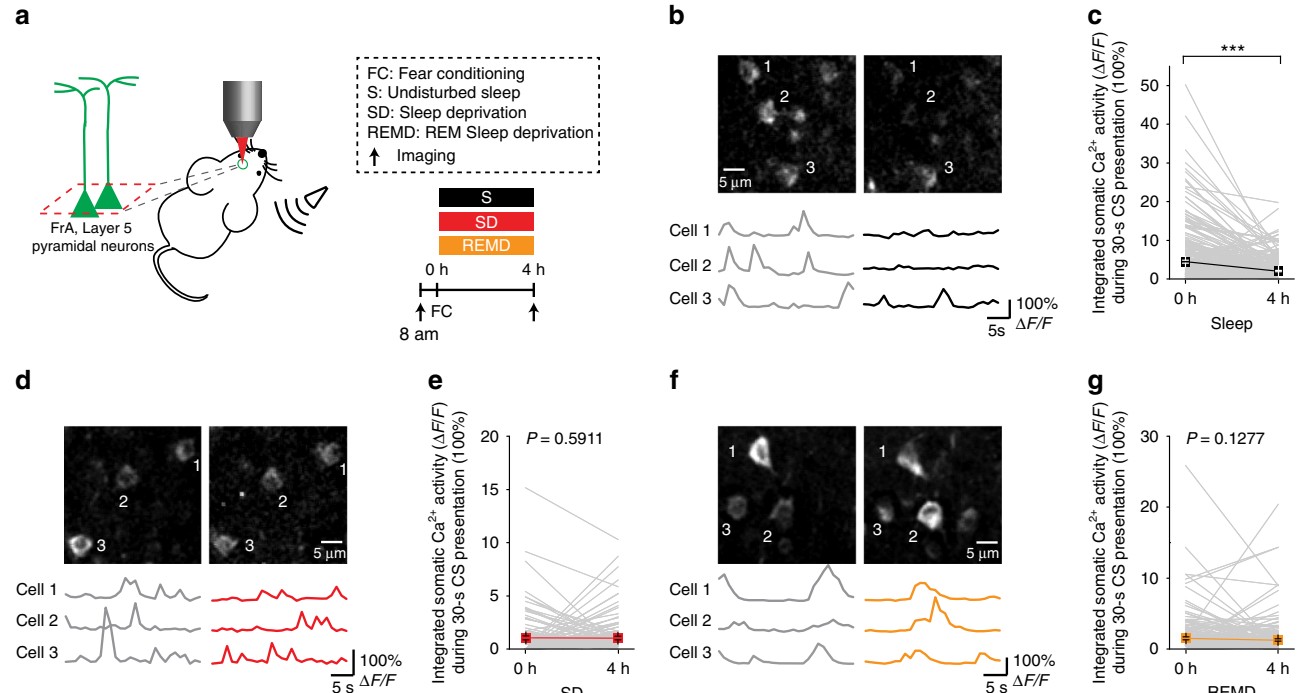

**Fig. 4 SD or REMD prevents FC-induced neuronal activity reduction. a** Schematic of experimental design. After fear conditioning, mice were either allowed to sleep or subjected to sleep/REM sleep deprivation. Two-photon $Ca^{2+}$ imaging was performed to assess the effect of sleep/REM sleep on somatic activity of layer 5 pyramidal neurons in FrA. **b** Somatic $Ca^{2+}$ activity of layer 5 pyramidal neurons in FrA before (left) and after (right) 4-h sleep. $\Delta F/F_0$ was summed to measure somatic $Ca^{2+}$ activity over 30 s. Scale bar: 5 μm. 3 traces: somatic $Ca^{2+}$ activities of cells on the top. $Ca^{2+}$ fluorescence traces over 30 s are shown. **c** The integrated somatic $Ca^{2+}$ activity during 30-s CS presentation was significantly decreased after FC and 4-h undisturbed sleep ($n = 309$ cells from 4 mice, $P < 0.0001$, Wilcoxon signed-rank test, two-sided). **d** Somatic $Ca^{2+}$ activity of layer 5 pyramidal neurons in FrA before (left) and after (right) 4-h SD. **e** SD for 4 h significantly prevented the reduction of neuronal activity ($n = 122$ cells from 5 mice, $P = 0.5911$, Wilcoxon signed-rank test, two-sided). **f** Somatic activity of layer 5 pyramidal neurons in FrA before (left) and after (right) 4 h of REMD. **g** REMD over 4 h significantly prevented the reduction of neuronal activity ($n = 195$ cells from 3 mice, $P = 0.1277$, Wilcoxon signed-rank test, two-sided). ***$P < 0.001$. Data are presented as mean ± s.e.m. Source data for (**c**), (**e**), and (**g**) are provided as a Source Data file.

significantly lower in SD mice than in mice with sleep, REMD or NREM-d, suggesting the role of NREM sleep in transient dendritic spine formation in FrA after FC. Taken together, these findings suggest that NREM and REM sleep have different roles in the rewiring of synaptic circuits in response to sensory and learning experiences. NREM is important for promoting the experience-dependent formation of new synapses of layer 5 pyramidal neurons, whereas REM sleep has a critical role in experience-dependent pruning of existing synaptic connections.

Our findings show that SD or REMD decreases MD-induced dendritic spine elimination and somatic activity reduction of layer 5 pyramidal neurons corresponding to the deprived eye during the critical period of mouse visual cortex development. It is possible that dendritic spines driven by the inputs from the deprived eye are preferentially eliminated, leading to reduced neuronal responses to visual stimuli. These results are consistent with previous findings in the developing cats that SD or REMD inhibits ocular dominance plasticity in which MD results in a shift of cortical responses to the non-deprived eye[15,35]. It is worth noting that in cats, sleep is necessary for the loss of responses to the deprived eye and gain of responses to the non-deprived eye, as measured by evoked extracellular recordings, kinase activity, and synaptic proteins[15,53–56]. On the other hand, in mice, the response to MD over hours is principally a loss of response to the deprived eye (contralateral visual cortex), followed by a much slower gain of response to the open eye. These differences may in part reflect differences in the underlying circuitry and experimental designs. Mice have a much stronger contralateral input (bias) whereas cats have a more balanced set of projections to the

visual cortex[57]. In addition, the present study examines dendritic spines and somatic activity of layer 5 pyramidal neurons, whereas earlier studies in cat measure evoked neuronal responses from all cortical layers[15,53–55].

In addition to MD-dependent changes in the developing visual cortex, REM sleep is important for FC-induced elimination of dendritic spines and reduction of somatic activities of layer 5 pyramidal neurons in FrA, which likely contribute to the increased freezing behavior. In humans, the recall after FC is enhanced after sleep, and conditioned response is positively correlated with the time spent in REM sleep[36]. REMD also inhibits the extinction of conditioned fear responses in humans[40]. Moreover, REMD impairs the consolidation of fear extinction memory in rats[41,42] and the attenuation of REM sleep theta rhythm in mice impairs fear contextual memory consolidation[43]. Together, these findings suggest that REM sleep has a general function in emotional memory consolidation in many species, including humans and rodents[36–43], potentially involving experience-dependent synapse elimination as demonstrated here. Interestingly, in contrast to the rapid effect on FC-induced spine elimination, REM sleep has no significant effect on spine elimination within the first 4 h MD, but after an initial 4 h MD. It is unclear what mechanisms underlie different time courses of REM sleep-induced spine elimination in FrA and V1. Because auditory cues paired with foot shocks during FC induce robust changes of neuronal activity in FrA whereas MD causes progressive changes of neuronal activity in V1 over an extended period of time (not shown), it is possible that the effect of REM sleep on experience-dependent synapse elimination depends on the level of neuronal activity triggered by the experience.

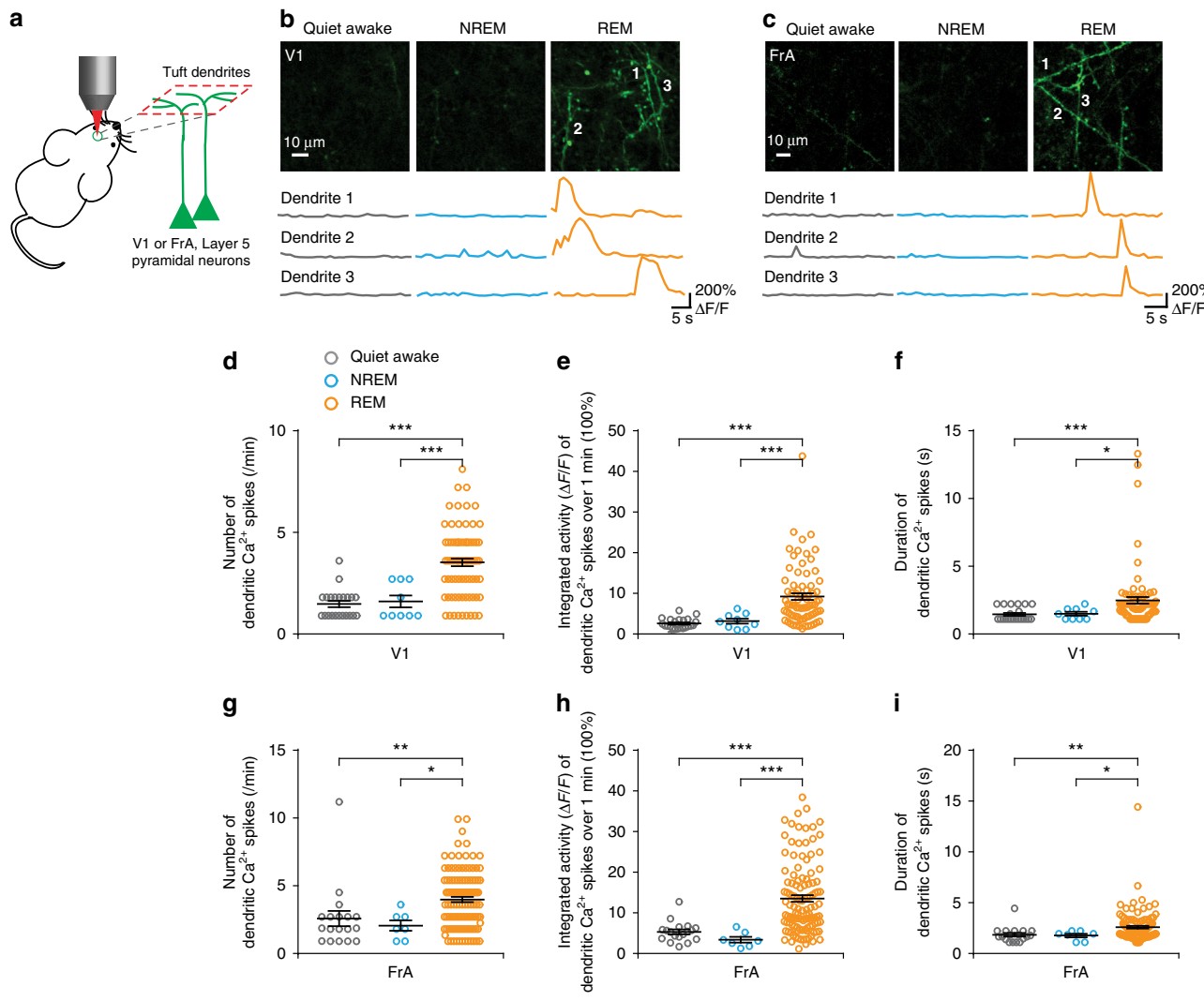

**Fig. 5 Dendritic Ca²⁺ spikes increase during REM sleep. a** Two-photon Ca²⁺ imaging of apical tuft dendrites of layer 5 pyramidal neurons in head-restrained mice during quiet awake, NREM sleep, and REM sleep. **b, c** Ca²⁺ imaging of apical tuft dendrites under various states in V1 (**b**) and FrA (**c**). Scale bar, 10 μm. Ca²⁺ fluorescence traces of three dendrites over 30 s are shown. **d, g** The number of dendritic Ca²⁺ spikes during REM sleep was significantly higher than that in other brain states in both V1 (**d**) (n = 4 mice, P < 0.0001 and = 0.0004 for REM sleep vs. quiet awake and NREM sleep, respectively) and FrA (**g**) (n = 5 mice, P = 0.0019 and = 0.0117 for REM sleep vs. quiet awake and NREM sleep, respectively) (Wilcoxon–Mann–Whitney test, two-sided). **e, h** The integrated activity of dendritic Ca²⁺ spikes during REM sleep was significantly higher than those during NREM sleep and quiet awake state in both V1 (**e**) (n = 4 mice, P < 0.0001 and = 0.0003 for REM sleep vs. quiet awake and NREM sleep, respectively) and FrA (**h**) (n = 5 mice, P < 0.0001 for REM sleep vs. quiet awake and NREM sleep, respectively) (Wilcoxon–Mann–Whitney test, two-sided). **f, i** The duration of dendritic Ca²⁺ spikes during REM sleep was significantly longer than those during other brain states in both V1 (**f**) (n = 4 mice, P = 0.0002 and = 0.0163 for REM sleep vs. quiet awake and NREM sleep, respectively) and FrA (**i**) (n = 5 mice, P = 0.0012 and = 0.0394 for REM sleep vs. quiet awake and NREM sleep, respectively) (Wilcoxon–Mann–Whitney test, two-sided). *P < 0.05, **P < 0.01, ***P < 0.001. Data are presented as mean ± s.e.m. In (**d–i**), each circle represents one active dendrite, n = 22, 9, and 78 dendrites for quiet awake, NREM sleep and REM sleep, respectively in V1; n = 18, 7, and 118 dendrites for quiet awake, NREM sleep and REM sleep, respectively in FrA. Source data for (**d–i**) are provided as a Source Data file.

It has been shown that dendritic Ca²⁺ spikes increase in the motor cortex during REM sleep and are involved in the pruning and strengthening of newly formed spines induced after motor training[19]. Consistently, our data show that the frequency, integrated activity, and duration of dendritic Ca²⁺ spikes are higher during REM sleep than during quiet wakefulness and NREM sleep in both V1 and FrA. Blockade of dendritic Ca²⁺ spikes during REM sleep with NMDA receptor antagonist MK-801 inhibits MD- and FC-induced dendritic spine elimination in V1 and FrA, respectively, while local puffing of MK-801 briefly during NREM sleep shows no effect on dendritic spine elimination. These results support the role of increased dendritic Ca²⁺ spikes during REM sleep in experience-dependent elimination of

existing connections. It is important to note that a recent study has shown that Ca²⁺ activity of apical shaft dendrites (200–400 μm below the pial surface) of layer 5 pyramidal neurons during REM sleep is comparable to that during NREM sleep but lower than the intermediate stage (a transitional sleep state) in the mouse somatosensory cortex[58]. In our study, elevated dendritic Ca²⁺ spikes during REM sleep were observed on layer 5 pyramidal apical tuft branches located within 60 μm below the pial surface. These findings raise the possibility that during REM sleep, dendritic Ca²⁺ spike-mediated spine elimination of layer 5 pyramidal neurons may occur predominantly in apical tuft branches located in the most superficial cortical layer 1. Future studies are needed to investigate how these dendritic Ca²⁺ spikes

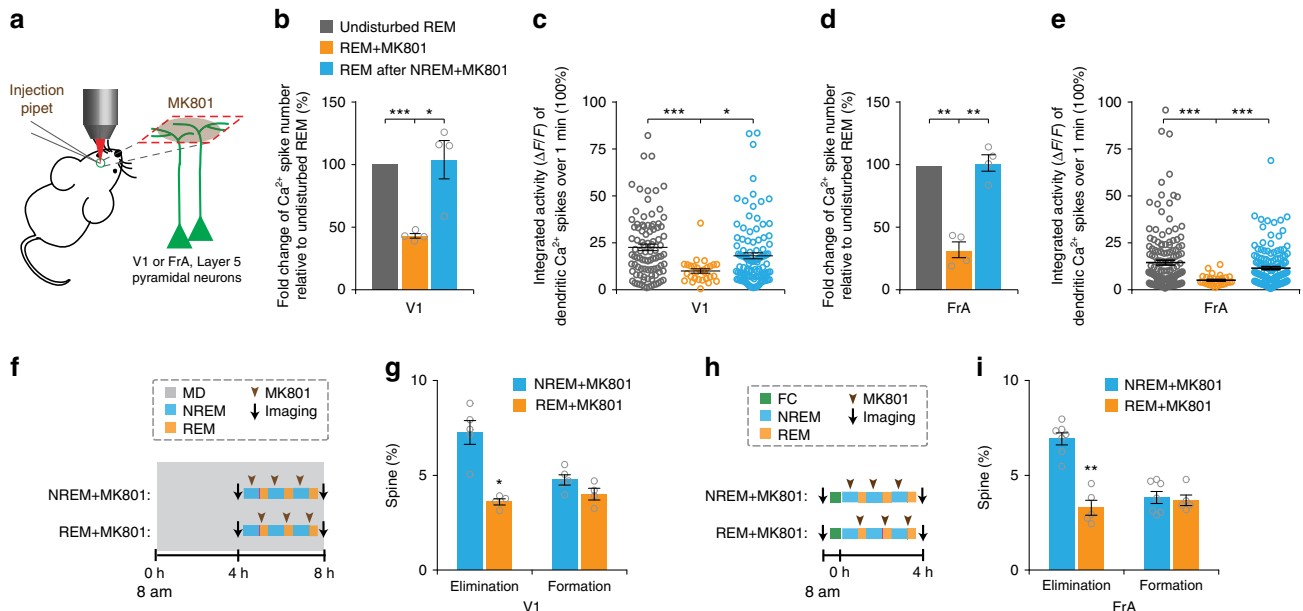

**Fig. 6 Blockade of dendritic Ca²⁺ spikes reduces spine elimination. a** Schematic of local injection of MK801 into the layer 1 of the mouse cortex under a two-photon microscope. **b, d** Local and brief injection of MK801 at the beginning of each REM sleep episode, but not during NREM sleep, significantly reduced the number of dendritic Ca²⁺ spikes during REM sleep as compared to that in undisturbed REM sleep in both V1 **b** ($n = 4$ mice, $P < 0.0001$ and $= 0.0362$ for REM + MK801 vs. undisturbed REM sleep and REM after NREM + MK801, respectively) and FrA **d** ($n = 4$ mice, $P = 0.0017$ and $= 0.00097$ for REM + MK801 vs. undisturbed REM sleep and REM after NREM + MK801, respectively) (paired $t$ test, two-sided). **c, e** MK801 injection at the beginning of a REM sleep episode, but not during NREM sleep, significantly blocked the integrated activity of dendritic Ca²⁺ spikes during REM sleep as compared to that in undisturbed REM sleep in both V1 **c** ($P = 0.0001$ and $= 0.0199$ for REM + MK801 vs. undisturbed REM sleep and REM after NREM + MK801, respectively) and FrA **e** ($P = 0.0003$ and $< 0.0001$ for REM + MK801 vs. undisturbed REM sleep and REM after NREM + MK801, respectively) (Wilcoxon–Mann–Whitney test, two-sided). **f** Schematic of experimental design to examine the effect of blocking dendritic Ca²⁺ spikes on spine elimination induced by MD in V1. **g** MD-induced dendritic spine elimination was significantly reduced after brief MK801 injection in REM sleep over a 4-h period as compared to that in NREM-d mice ($n = 4$ mice for REM + MK801; $n = 5$ mice for NREM + MK801, $P = 0.0159$, Wilcoxon–Mann–Whitney test, two-sided). **h** Schematic of experimental design to examine the effect of blocking dendritic Ca²⁺ spikes on spine elimination induced after FC in FrA. **i** Brief injection of MK801 during REM sleep, but not during NREM sleep, reduced FC-induced dendritic spine elimination ($n = 5$ mice for REM + MK801; $n = 7$ mice for NREM + MK801, $P = 0.0025$, Wilcoxon–Mann–Whitney test, two-sided). *$P < 0.05$, **$P < 0.01$ ***$P < 0.001$. Data are presented as mean ± s.e.m. In (**c**) and (**e**), each circle represents one active dendrite, $n = 29$, 94, and 108 dendrites for REM + MK801, undisturbed REM sleep and REM after NREM + MK801, respectively in V1; $n = 32$, 158 and 144 dendrites for REM + MK801, undisturbed REM sleep and REM after NREM + MK801, respectively in FrA. Source data for (**b–e**), (**g**) and (**i**) are provided as a Source Data file.

are generated during REM sleep (e.g., if back-propagating action potentials are involved) and whether they may cause spine elimination of dendritic branches located in deeper cortical layers.

The precise mechanisms underlying Ca²⁺ spike-dependent spine elimination also remain to be determined. Dendritic Ca²⁺ spikes have been shown to cause potentiation and de-potentiation depending on the time window of synaptic activity and dendritic spike generation[44–49]. The timing between dendritic Ca²⁺ spikes and spine activity during REM sleep could lead to activation of signaling pathways important for dendritic spine elimination in V1 and FrA. It has been shown that extracellular signal-regulated kinase (ERK) phosphorylation and the expression of immediate-early genes such as Zif268 increase during REM sleep[15,17,55]. Because REM sleep duration is short (~1–2 min in mice) and spine elimination does not seem to occur during the brief REM sleep episode (unpublished observations), it is likely that dendritic Ca²⁺ spikes during REM sleep set in motion downstream Ca²⁺ signaling cascades that could facilitate the elimination of existing spines triggered by experiences. Furthermore, it is important to note that MK801 blockade of NMDA receptors could influence spine elimination not only by reducing dendritic Ca²⁺ spikes but also via non-ionotropic NMDA receptor signaling[59,60]. Further studies are needed to examine the generation of dendritic Ca²⁺ spikes and down-stream signaling pathways to further delineate

the role of REM sleep in modulating synaptic connectivity in development and learning.

## Methods

**Experimental animals.** *Thy1*-YFP-H transgenic mice expressing YFP in layer 5 pyramidal cells were group housed (5 per cage) either in the animal facility at New York University School of Medicine or in the Peking University Shenzhen Graduate School (PKUSZ) or the University of Hong Kong. Transgenic mice expressing GCaMP6s in pyramidal neurons (*Thy1.2*-GCaMP6s Line 1[61]) were group housed (5 per cage) in the animal facility at New York University School of Medicine. Mice were maintained at 22 ± 2 °C and humidity-controlled room with a 12-h light: dark cycle (lights on at 6:30 am, lights off at 6:30 pm). All experiments were conducted during the light cycle, starting around 08:00. Food and water were available ad libitum. Young mice (postnatal day 27–30) of both sexes were used in this study. All animal procedures were performed in accordance with protocols approved by the Institutional Animal Care and Use Committee of New York University School of Medicine, PKUSZ and the University of Hong Kong.

**Auditory-cued FC and MD.** For FC, mice were trained and tested using the FreezeFrame system (Coulbourn Instruments). Behavior was recorded using a low-light video camera. Stimulus presentation was automated using Actimetrics FreezeFrame software (version 2.2; Coulbourn Instruments). All equipment was thoroughly cleaned with detergent followed by water between sessions. Mice were habituated for 2 min on a shocking grid (cage set-up A: shocking floor grids, ethanol scent). Fear conditioning was conducted with three pairings of a 30-s, 4000-Hz, 80-dB auditory cue (CS) co-terminating with a 2-s, 0.5-mA scrambled footshock (US). The inter-trial interval was 15 s. Two minutes after conditioning, mice were returned to their home cages. For the unpaired control group, mice

received tones and shocks in an unpaired manner (tones and shocks were separated by random intervals of 5–15 s). Mice were returned to their home cages 2 min after the presentation of the unpaired stimuli. For the recall test, mice were placed in a different context (cage set-up B: test floor grids, 1% Pinesol) for an initial 2-min period and then subjected to tone (CS) presentation for 2 min.

For MD, the contralateral eye of the animal was covered by a shade cloth, which is in dark color and lightproof. The shade cloth is about 1.5 mm × 1.5 mm in size; its edge is attached to the dental cement. Animals were returned to the home cage and under lighted condition between imaging sessions.

### Surgery for imaging and EEG/EMG recording. 
Head mount/EEG implantation, skull thinning surgery or a small craniotomy window surgery were performed in anesthetized mice according to previously published studies[11,19]. Specifically, 24 h before imaging, surgery was performed to attach a head holder and to create a thin-skull cranial window or to perform a small craniotomy (~200 μm) and replacement of the skull with a cover glass in mice deeply anesthetized with an intraperitoneal injection of ketamine (100 mg/kg) and xylazine (10 mg/kg). The mouse head was shaved and the skull surface was exposed with a midline scalp incision. The periosteum tissue over the skull surface was removed without damaging the temporal and occipital muscles. A head holder composed of two parallel micro-metal bars was attached to the animal's skull for head restraint to reduce motion-related artifacts during imaging. A small skull region (~0.2 mm in diameter) over the FrA[29] (2.8 mm anterior from bregma and 1.0 mm lateral from the midline) or the binocular region of primary visual cortex[30] (3.0 mm posterior from bregma and 3.0 mm lateral from the midline) was identified based on stereotaxic coordinates and marked with a pencil. A thin layer of cyanoacrylate-based glue was first applied to the top of entire skull surface, and then the head holder was mounted on top of the skull with dental acrylic cement such that the marked skull region was exposed between the two bars. Precaution was taken not to cover the marked region with dental acrylic cement.

In fear conditioning experiments, four electrodes were implanted to allow simultaneous imaging and EEG/EMG recording in the same animal. Two electrodes were used for recording epidural EEG and two for recording EMG. Each electrode was made by soldering one end of an epoxy coated silver wire (0.005 inches in diameter, A-M Systems) to a connector pin. One EEG electrode was placed over the left frontal cortex (2 mm lateral to midline, 2 mm anterior to bregma) and another on the cerebellum (at midline, 1 mm posterior of lambdoid suture). Before the electrode implantation, a small area of the skull (each ~0.2 mm in diameter) was thinned with a high-speed drill and carefully removed with forceps. The electrodes were bent at 1 mm from the tip of the silver wire and carefully inserted under the skull above the dura mater. The electrodes were fixed by cyanoacrylate-based glue and further stabilized by dental cement. Two electrodes for EMG recording were placed on the nuchal muscle and stabilized, together with the EEG electrodes, with dental cement.

In MD experiments, the EEG electrodes were inserted above the right hemisphere of the visual cortex ipsilateral to the deprived eye (3 mm lateral to midline, 3 mm posterior to bregma). Before the implantation, a small area of the skull was first thinned with a high-speed drill, and two small pieces of skull (~0.2 mm) was carefully removed with forceps. Two electrodes for EEG recording were made by soldering one end of an epoxy-coated silver wire (0.003 inch in diameter, A-M Systems) to a connector pin. The epoxy at the other end of the electrodes was removed and the exposed silver wires were carefully inserted into the superficial layer of the cortex (~100 μm below the pial surface) with the tips separated from each other by ~800 μm. The electrodes were fixed first by cyanoacrylate-based glue and subsequently by dental cement. Two electrodes for EMG recording were made of polyurethane enameled copper wires (0.13 mm in diameter), placed on the nuchal muscle and stabilized, together with the EEG electrodes, with dental cement.

After the dental cement was completely dry, the head holder was screwed to two metal cubes attached to a solid metal base, and an imaging window was created over the previously marked region. A small skull region ~1 mm in diameter over the FrA[29] or the binocular region of primary visual cortex[30] (based on stereotaxic coordinates) was thinned with a high-speed drill in order to create a thinned-skull or craniotomy window. In brief, a high-speed drill was used to carefully reduce the skull thickness by ~50% under a dissecting microscope. Skull thinning was completed by carefully scraping the cranial surface with a microsurgical blade to create a region ~20 μm in thickness and ~200 μm in diameter.

For imaging dendritic structure and activity, a thin layer of silicon was applied to cover the thinned-skull window after the surgery. The animals were returned to their own cages to recover.

For imaging somatic Ca²⁺ of layer 5 pyramidal neurons, a small craniotomy window (~200 μm in diameter) was made and immediately covered with a glass coverslip in anesthetized mice. The coverslip was glued to the skull to cover the exposed cortex and reduce brain motion. Twenty-four hours later, somatic Ca²⁺ activity of layer 5 pyramidal neurons was imaged under awake and head-fixed condition twice, each lasting for ~30 min. It is worth noting that we imaged somatic Ca²⁺ of layer 5 pyramidal neurons through a cranial window (made 24 h before imaging), rather than a thinned skull window or allowing 1–2 weeks recovery after cranial window surgery. This was due to the difficulty to image somas of layer 5 pyramidal neurons located >500 μm through a thinned skull.

We found that cranial window surgery over a large region (5 mm in diameter) would cause inflammation that was sometimes mitigated after 1–2 weeks of recovery. In our hands, most of the mice undergoing such cranial window surgery still could not be imaged after a 1–2 weeks recovery. On the other hand, we were able to image somas of pyramidal neurons in layer 5 in most animals if a small cranial window (~200 μm) was made and covered by a coverglass 24 h (rather than 1–2 weeks) before imaging.

Before two-photon imaging start, mice were given at least 24 h to recover from the surgery-related anesthesia (head mount/EEG and EMG implantation, skull thinning or a small craniotomy window surgery) and habituated for a few times (10 min each) under the imaging apparatus to minimize potential stress effects due to head restraining and awake imaging.

### SD procedure. 
SD was achieved through gentle handling over a period of 4 h after fear conditioning or during MD under lighted condition. Specifically, mice were gently touched with a cotton applicator for 1–2 s based on EEG and EMG recordings whenever they displayed signs of slow-wave sleep. Food and water were available ad libitum throughout the entire deprivation process.

### REM SD and NREM sleep disturbance procedure. 
REM SD was achieved through gentle handling over a period of 4 h after FC or during 4–8 h MD under lighted condition. Specifically, mice were gently touched with a cotton applicator for 1–2 s as soon as they displayed signs of REM sleep based on EEG/EMG recordings[19]. The EEG/EMG trace was scored by an observer in real time. On average, when EEG and EMG traces start to show a sign of REM sleep around 6–8 s, animals were gently touched. On average, mice were touched ~2 or 3 times per hour at ~P27–30 during the period of REM SD. The same number of touches was then used during NREM sleep to control for the disturbance caused by gentle handling. The animals were not previously habituated to this gentle handling protocol. Food and water were available ad libitum throughout the entire deprivation process.

### MK801 application. 
For local application of MK801 (200 μM in artificial cerebrospinal fluid (ACSF), M107, Sigma-Aldrich), mice were head-restrained for 4 h and a glass microelectrode (~20 μm outer diameter) was inserted through a bone flap into the superficial layer of the cortex (~60 μm below the pial surface) with an angle of 30° toward and ~100 μm away from the imaging area[19]. The bone flap (~50 μm in diameter) was made adjacent to a thinned skull window, 24 h before drug delivery in anesthetized mice. The area for MK801 injection and imaging was covered with ACSF. Immediately upon the detection of REM sleep based on EEG and EMG recordings, MK801 (200 μM in ACSF, ~40 nl) was injected via pressure injections with Picospritzer II (40 p.s.i., 30 ms per pulse, 1 Hz, ~3–4 pulses; General Valve Corporation) into the FrA or V1 of head-fixed mice. At the same time, Ca²⁺ imaging of dendritic activities was performed in superficial layer of the FrA and V1. The maximum duration of the imaging/recording session during sleep was 4 h.

### Two-photon Ca²⁺ imaging in mice expressing GCaMP6s. 
Transgenic mice expressing genetically encoded Ca²⁺ indicator GCaMP6s were used for Ca²⁺ imaging of apical dendrites and somas of layer 5 pyramidal neurons (depth 520–650 μm) in the primary visual cortex or FrA. Mice were given at least 24 h to recover from the surgery-related anesthesia before imaging. Prior to imaging, mice with head mounts were habituated for a few times (10 min each time) in the imaging apparatus to minimize potential stress effects of head restraining.

For imaging Ca²⁺ activities of dendrites (~60 μm below the pial surface), mice were head-restrained in the imaging apparatus and allowed to sleep under the two-photon microscope with a heating pad (temperature controller 35–36 °C) attached to the base of the imaging apparatus throughout the imaging. Ca²⁺ signals were recorded during NREM sleep and REM sleep (3–5 times, 1–2 min every time) based on EEG/EMG recording. Dendritic Ca²⁺ activities were recorded through a thinned-skull window with the overlaying silicon being removed. Two-photon imaging was performed with an Olympus Fluoview 1000 two-photon system equipped with a Ti:Sapphire laser (MaiTai DeepSee, Spectra Physics) tuned to 920 nm. The average laser power on the tissue sample was ~15 mW for imaging the superficial layer of the cortex. All experiments for imaging dendritic Ca²⁺ signal under various brain states were performed with a 25× objective at ~1 Hz (N.A. 1.05, 3× digital zoom). Experiments for MK801 injection during REM sleep or NREM sleep were performed using a 40× objective (N.A. 0.8, 3× digital zoom) at ~1 Hz. Image acquisition was performed using FV10-ASW v.3.0 software.

We performed somatic Ca²⁺ imaging at the depth of ~550 μm below the pial surface through a small craniotomy window (~200 μm in diameter) in mice under awake and head-fixed conditions twice, each lasting for ~30 min. In FC experiments, mice were placed under the two-photon microscope and imaged during the presentation of a 4000 Hz auditory tone over 30 s. The laser was tuned to the wavelength of 920 nm with laser power of ~30 mW on the tissue sample. Ca²⁺ signals were recorded at 1 Hz using a 25× objective (N.A. 1.05, 1.5× digital zoom).

In MD experiments, round-shaped and square-wave black/white drifting gratings (0.08 cycles per degree, 4 cycles per second, covering 32° × 32° screen area as seen by the mouse) of changing orientations (8 directions) were used as visual

stimuli[30]. The stimuli were only presented to the contralateral eye (deprived) in front of the central visual field of the mouse with the ipsilateral eye covered by a shade cloth (75 Hz refresh rate of the monitor, 18 cm away from the mouse's contralateral eye). $Ca^{2+}$ imaging was performed at 2 Hz using a 25× objective (N.A. 1.05, 1.2× digital zoom) over a 1-min period of visual stimulation.

**Imaging dendritic spine remodeling in awake mice**. Image stacks of dendritic segments projecting to superficial cortical layers were obtained using an Olympus two-photon microscope (FV1000MPE) with the laser tuned to 920 nm. The images were acquired using 1.10 N.A. 60× objective or 1.05 N.A. 25× objective immersed in ACSF. A low magnification stack (200 μm × 200 μm; 512 pixel × 512 pixels for 60 objective, 2 μm Z-step size; 508 μm × 508 μm; 1024 pixel × 1024 pixel for 25× objective, 2 μm Z-step size) of fluorescently labeled neuronal processes was taken and used as a map for the relocation of the same area at later time points, in addition to the marked brain vasculature map. Two to three stacks of image planes (66.7 μm × 66.7 μm, 512 pixel × 512 pixels for 60× objectives, 0.75 μm Z-step size; 169 μm × 169 μm, 1024 pixel × 1024 pixel for 25× objectives, 0.75 μm Z-step size) within a depth of 60 μm from the pial surface were collected at each time point, yielding a full three-dimensional data set of dendrites in the area of interest. The animal was head restrained during image acquisition that took ~30 min, and immediately released to its original cage and stayed there until the next imaging sessions. The window for imaging was covered with a thin layer of silicon or a drop of 1% agar (<40 °C) to prevent drying between imaging sessions.

**Data analysis of spine structural plasticity**. All data analysis was performed blindly to treatment conditions and all dendrites were chosen for analysis as long as these dendrites exhibited good signal to noise ratio and did not overlap with dendrites nearby. The procedure for quantifying spine dynamics has been described in the earlier studies[11,19,29,30]. In brief, image stacks were analysed using NIH Image J software. For each dendritic segment analysed, filopodia were identified typically as long, thin protrusions with ratio of head diameter to neck diameter <1.2:1 and ratio of length to neck diameter >3:1. Short, thin protrusions with the ratio of head diameter to adjacent dendritic shaft diameter <1:2 and ratio of head intensity to adjacent dendritic shaft intensity <1:3 were also identified as filopodia. The remaining protrusions were classified as spines. Such criteria are based on previously published work[11,62]. We distinguished spines versus filopodia because filopodia are highly dynamic/transient structures and previous studies have shown MD or FC over days did not cause changes in filopodial dynamics[29,30].

Spines were considered the same between views if their positions remained the same distance from relative adjacent landmarks. Spines were considered different if they were more than 0.7 μm away from their expected positions based on the first view. More than 150 spines were analysed from each animal. The minimum length of the dendritic branches included in the analysis was ~30 μm. The degree of spine elimination or formation was calculated as the number of spines eliminated or added divided by the number of pre-existing spines. For dendritic spine image display, surrounding fluorescent structures that were not related to the dendrites of interest were removed manually from image stacks using Adobe Photoshop. This procedure did not alter the data or data quantification itself (Supplementary movies 1–4). The modified image stacks were then projected to generate two-dimensional images and adjusted for contrast and brightness.

**Somatic and dendritic $Ca^{2+}$ imaging data analysis**. For somatic and dendritic $Ca^{2+}$ analysis, we quantified $Ca^{2+}$ activity of somas and dendrites (>30 μm in length) blind to various brain states and treatment conditions. Neuronal $Ca^{2+}$ activity, indicated by GCaMP6 fluorescence changes, was analyzed post hoc using ImageJ software (NIH) according to previous studies[30,44]. All imaging stacks were registered using ImageJ plugin StackReg. The GCaMP6 fluorescence ($F$) during visual stimulation or tone presentation was measured by averaging pixels within each visually identifiable soma (Regions of interests, ROIs). Changes of fluorescence $\Delta F/F_0$ was calculated as $\Delta F/F_0 = (F - F_0)/F_0$, in which $\Delta F$ was $F - F_0$ (all $F$ values were subtracted from a background fluorescence) and $F_0$ was the average of 10% minimum $F$ values over 1 min or 30 s period, representing baseline fluorescence. $\Delta F/F_0$ above three times the standard deviation (SD) of baseline fluorescence was summed to measure somatic calcium activity of layer 5 neurons over 1 min or 30 s. In Supplementary Figs. 4, 6, change in the integrated $Ca^{2+}$ activity of individual cells was calculated by subtraction from the second to the first imaging session.

Dendritic $Ca^{2+}$ activities during the quiet awake state, NREM, and REM sleep, as indicated by GCaMP6s fluorescence changes, were also analysed post hoc using ImageJ software (NIH). ROIs corresponding to visually identifiable apical tuft dendrites were selected for quantification. On average, we quantified $Ca^{2+}$ activity on dendritic segments ~30 μm in length. As described in previous studies[19,44], dendritic $Ca^{2+}$ spikes were defined as the events when changes of fluorescence ($\Delta F/F_0$) observed in both dendritic spines and shaft (average length >30 μm) were >50% for GCaMP6s during the imaging sessions. $F_0$ is the fluorescence intensity in dendritic segments after background subtraction. The threshold for detecting dendritic spikes was >3 × SD of baseline fluorescence noise for GCaMP6s. The majority of $Ca^{2+}$ spikes were found to have a fluorescence increase of 100 to >1000% for GCaMP6s. $\Delta F/F_0$ above threshold was summed to measure the integrated activity of dendritic $Ca^{2+}$ spikes over 1 min. The full width of an individual spike was measured as the spike duration. The number, duration, and integrated activity of $Ca^{2+}$ spikes were quantified from individual $Ca^{2+}$ spikes that did not overlap with other spikes under various brain states.

**EEG/EMG recording and analysis**. Between imaging sessions, EEG/EMG was recorded using the BL-420F Biological Data Acquisition & Analysis System (Chengdu TME Technology Co., Ltd, China) with a bandpass setting of 0.1–100 Hz. EEG/EMG data were visually scored for awake and sleep states[19]. Wakefulness was identified by low amplitude and high-frequency EEG activity and high EMG activity. REM sleep was identified by low amplitude and high power at theta frequency (5–9 Hz) EEG activity and low EMG activity. NREM sleep was identified by high amplitude and low frequency (0.5–4 Hz) EEG activity and low EMG activity. To verify our on-line visual scoring of the animal's state, we performed posthoc analysis. We used 10 s episodes of EEG recordings. Each episode was low pass filtered at 40 Hz and then Fast Fourier Transformed to convert EEG waveform from the time domain to the frequency domain. We normalized the power spectrum density by the average power over all frequencies (0–40 Hz). All REM sleep periods 5 s or longer were analyzed. REM sleep occupied ~4–6% of the total time (including sleep and wake) in non-deprived mice or NREM-d mice within 4 h at P27 - 30.

**Statistics and reproducibility**. All data are presented as mean ± s.e.m. Sample sizes were chosen to ensure adequate power with the statistical tests while minimizing the number of animals used in compliance with ethical guidelines. We used Wilcoxon–Mann–Whitney test, Wilcoxon signed-rank test or paired $t$ test to compare two groups, and Kruskal–Wallis test to compare more than two groups. Kruskal–Wallis test was followed by Tukey-Kramer or Turkey LSD or Dunn's test for multiple comparisons. All tests were conducted as two-sided tests. Significant levels were set at $P \leq 0.05$. All statistical details of the experiments can be found in the figure legends. Figures 1a; 2b, d, f; 3a; 4b, d, f; 5b, c are images of dendrites or cells from individual animals. Each experiment was repeated independently with similar results that were quantified and presented in corresponding figures.

**Reporting summary**. Further information on research design is available in the Nature Research Reporting Summary linked to this article.

## Data availability
Source data for Figs. 1b, c, e, g; 2c, e, g; 3b–d, f, g; 4c, e, g; 5d–i; 6b-e, g, i and Supplementary Figs. 1a–f, 2a–d, 3b, 4, 5b, 6, 8 are provided in a Source Data file. All data that support the findings of this study are available from the corresponding author upon reasonable request.

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

## Acknowledgements

We thank Dr. Avital Adler for helping MATLAB analysis of EEG and EMG patterns. We also thank all the members in the Gan laboratory for comments on the manuscript. This study was supported by NSFC (81771428) to W.L., Shenzhen Science and Technology Innovation Funds (JCYJ 20180302150304250, JCYJ20170807144417251 and JCYJ20180504165435326) to Y.B. and Y.M.Z., RGC/ECS (27103715), RGC/GRF (17128816), NSFC (31571031) to C.S.W.L., and NIH R35 GM131765 to G.Y. and R01 NS047325 to W.-B.G.

## Author contributions

Y.M.Z., C.S.W.L., M.G.F., and W.-B.G. designed the experiments. Y.M.Z. performed dendritic structure imaging, somatic activity imaging in MD experiments and analyzed the data with the help from Y.B. and W.-B.G. C.S.W.L. performed dendritic structure

imaging and behavioral tests in FC experiments and analyzed the data with the help from G.Y. and W.-B.G., Y.M.Z. and Y.B. performed somatic activity imaging in FC experiments and data analysis. Y.M.Z. performed dendritic calcium imaging and dendritic structure imaging with MK801 injection experiments and analyzed the data with the help from W.L. and R.H.Z., Y.M.Z., C.S.W.L. and W.-B.G. prepared the manuscript with inputs from M.G.F.

## Competing interests

The authors declare no competing interests.
