## [Peer Review File · Nature Communications]

Reviewers' comments:

Reviewer #1 (Remarks to the Author):

In this manuscript the authors investigate the relationship between sleep and synapse elimination in the mouse primary visual cortex and in the frontal association cortex. They induce plastic changes using the powerful monocular deprivation approach as well as the robust auditory-cued fear conditioning approach. They report that both monocular deprivation and fear conditioning caused dendritic spine elimination. Importantly spine elimination was reduced after total sleep deprivation or REM sleep deprivation. The changes in spine elimination were associated with changes in layer5 pyramidal neuronal activity. Importantly, the authors observe an increase in dendritic calcium spikes during REM sleep and that blocking these spikes specifically during REM sleep prevents monocular deprivation and fear conditioning spine elimination.

This is a very interesting paper that explores a topic that is important to the field of sleep research specifically and neuronal plasticity more generally. The authors present a compelling case to support their hypotheses. The experiments are solid and the interpretation is, generally, in keeping with the data. It is important to note that the results provide a coherent and consistent picture. I am very enthusiastic about this data. This manuscript fills a hole in the field and is impressive. I do not have any major concerns.

Reviewer #2 (Remarks to the Author):

Overall it is very interesting, well done and important study. Here, using two-photon imaging, Y. Zhou et al demonstrate that high levels of Ca activity observed during REM sleep in one-month old mice contribute/lead to elimination of dendritic spines in visual and prefrontal cortices. The results are convincing. Despite my very positive evaluation of the study, I have several suggestions that will improve the presentation of findings.

Major.

1. Both, title and abstract do not contain any mentioning that the study was done during critical period of cortical development. Therefore, it might provide misleading information to a not careful reader.

2. The result section does not contain much of numerical info. In each section of results, please indicate how many of animals, dendrites or spines were used to obtain each piece of data.

Minor.

1. Videos. Add calibration bar throughout recording (not just on the first frame). Add time indicator. If possible, play it slower that the reader can clearly see activity. It appears that NREM is very brief in both videos (order of 10 sec if the movie was 40 sec). Could you show more convenient segments?

2. Fig. 6 c, e. Please indicate units.

3. Indicate injected volume of MK801.

4. Bone thinning described twice in the method section.

Reviewer #3 (Remarks to the Author):

In this manuscript, the authors use in vivo structural and functional imaging to show that REM sleep deprivation, or blockade of NMDARs during REM sleep, alters spine remodeling in the context of monocular deprivation and fear conditioning in visual (V1) and prefrontal cortices (FC), respectively. The results complement prior work on the effects of sleep deprivation in various sensory and behavioral paradigms, including prior work from the same group showing that REM

sleep deprivation or blockade of NMDARs during REM sleep alters motor learning-induced spine remodeling (Li et al., Nat. Neurosci. 2017). There are several reasons why this study is inappropriate for Nature Communications. First and foremost is the limited novelty. The work is incremental, at best, in relation to previously published work from the same group and provides no further conceptual insight. One of the main claims, repeated in both the abstract, results, and discussion, that blockade of dendritic calcium spikes specifically during REM sleep prevents MD- and FC-induced dendritic spine elimination in V1 and FC, is unsubstantiated (more below). Finally, there are major concerns related to the rigor of experimental procedures and the authors' transparency about their experimental procedures, as well as animal welfare. We outline these and other concerns below.

Major concerns:

This study primarily extends the authors' own previously published work from motor cortex (Li et al., Nat. Neurosci. 2017) to two additional cortical regions and behavioral/sensory paradigms. The authors over represent its novelty in their Introduction and Discussion by referring to their previous work in terms of spine maintenance and the current work in terms of spine elimination (Introduction, lines 77-79; Discussion, lines 255-256). The authors' statement that "the role of sleep in experience-dependent elimination of existing synapses has not been examined previously" (lines 255-256) is untrue, since the authors' own published work shows effects of REM sleep deprivation and MK-801 treatment on spine elimination in the context of motor learning (Li et al., Nat. Neurosci. 2017). In fact, their 2017 paper distinguishes elimination of pre-existing spines from elimination of new spines, while the current study cannot make such distinctions due to the use of only 2 imaging sessions for most experiments.

The MK-801 effects in Fig. 6 suggests that NMDARs mediate the effect of REM sleep on Ca spikes and experience-dependent spine remodeling. However, it is not possible to conclude that this is sequential, where NMDAR blockade eliminates Ca spikes and that results in reduced spine elimination during REM sleep. An alternative interpretation is that the MK801 treatment, by blocking NMDARs independently influences both Ca spikes and spine elimination. Indeed, as stated in the last paragraph of the discussion "Further studies are needed to examine the generation of dendritic Ca²⁺ spikes and down-stream signaling pathways to further delineate the role of REM sleep in modulating synaptic connectivity in development and learning."

The experimental design in most of the experiments lack important baseline controls. In the same way that supplementary fig. 3 shows controls for Figs 2&3 - a 4-hour imaging interval with sleep manipulation but without MD, there should be an equivalent control for Figs 3&4- prefrontal cortex without fear conditioning. Same for Fig 6, a control experiment is warranted testing whether MK-801 during REM sleep reduces spine elimination under baseline conditions, in animals that are not undergoing monocular deprivation or fear conditioning.

We question whether some of the experimental timelines described in this manuscript received appropriate animal welfare oversight. This is important both in terms of animal welfare, but also in terms of experimental confounds:

- The authors state that for somatic calcium imaging during sleep and awake states, "the skull overlying the cortex of interest was removed and replaced with a glass window right before two-photon imaging" (Methods, lines 493-494). It is unclear why the authors would perform cranial window surgery immediately before imaging, since cranial windows are routinely imaged for days or weeks after surgery (when using a glass implant, as the authors did for somatic calcium imaging). This is particularly concerning for an experiment involving functional imaging in awake animals, since recent major surgery under anesthesia may affect neuronal activity and responsiveness to the visual and auditory cues presented during imaging. Further, the authors themselves have previously published work showing inflammation immediately post cranial window surgery (Xu et al., Nat. Neurosci. 2007) that is only mitigated after a 1-2 weeks recovery

(Holtmaat et al., Nat. Protocols, 2009). This major confound could be avoided simply by giving the animals a recovery period after cranial window surgery.

- This design also raises concern about the welfare of animals undergoing prolonged experiments when the somatic calcium imaging immediately follows the window surgery, involving 2 awake head-fixed imaging sessions on the same day, with fear conditioning and 4 hours of sleep deprivation occurring between the imaging sessions. This is a demanding protocol even for mice that are not recovering from surgery. The authors' description of the timeline of surgery/anesthesia is vague, with only brief mention of the cranial window implantation in a section that focuses primarily on the imaging procedures. The timelines for implanting a head post (for head fixation to the microscope) and habituation to awake head fixation are also not described for the experiments shown in Fig. 2 and Fig. 4. This needs clarification and confirmation that this experimental timeline, including the lack of recovery time from cranial window surgery, was reviewed by the relevant institutional committee(s) on animal care. We see no scientific justification for the lack of surgery recovery time and strongly encourage the authors to give their mice appropriate recovery times in the future.
- The MK-801 experimental timeline shown in Fig. 6f and Fig. 6h suggests that animals were head-fixed (to enable MK-801 application via the Picospritzer) for the entire 4 hrs between imaging sessions, in addition to being head-fixed during the 2 imaging sessions. Please clarify in the Methods whether the animals remained head-fixed for the entire 4-hour period between imaging sessions, and confirm that the duration of head fixation was reviewed by the relevant institutional committee(s) on animal care. We do understand the scientific justification for this prolonged head fixation, i.e. enabling use of a Picospritzer that would be infeasible in freely moving animals. However, the authors should be transparent in their Methods about the duration of head fixation. We also encourage the authors to consider in their future studies whether providing a lick spout with water and/or calorific liquid would be appropriate during prolonged head fixation.

Minor concerns:

The authors state that they used Photoshop to remove "fluorescent structures near and out of the focal plane" in their example images (Methods, lines 517-518). We understand the difficulty of displaying a single dendrite from a 3D image stack in a 2D example image, but the use of Photoshop to remove fluorescent structures does not comply with Nature Communications' current image processing guidelines (<https://www.nature.com/nature-research/editorial-policies/image-integrity>). We request that the authors produce example images that comply with current image processing guidelines.

For calcium imaging of dendrites, the authors should clarify how motion artifacts were accounted for, since they did not use a cell fill to visualize dendrites during times of low Ca²⁺/GCaMP signal, and even slight physiological motion can cause dendrites to move out of focus in Z. The authors' use of StackReg for image registration (Methods, line 549) suggests that there were XY motion artifacts in their images, but StackReg cannot correct Z movement. In the supplemental videos, there are dendrite segments whose GCaMP fluorescence is only visible during Ca²⁺ spikes, and it is not possible to visually determine whether the dendrites are present with low Ca²⁺ or have moved out of focus when they are not visible in the videos. This issue is particularly concerning when calcium spikes are being compared between different sleep/awake states, since motion artifacts may differ between different sleep/awake states.

The authors describe their dendritic calcium events as dendritic spikes, but back-propagating action potentials (bAPs) are not distinguishable from dendritic spikes in this imaging paradigm. The MK-801 effects in Fig. 6b-e suggest that at least some of the Ca²⁺ events were dendritic spikes dependent on excitatory synaptic input in L1, but the authors should acknowledge in the text of the manuscript that some of the dendritic Ca²⁺ events they observe could originate from bAPs rather than dendritic spikes.

For somatic calcium activity (Fig. 2 and Fig. 4), please clarify in the legend the measure used on

the Y-axis (e.g. is this the number of events, or the percent of CS presentations that produced an event, etc).

For the data shown in Fig. 2 and Fig. 4, the authors should provide a statistical comparison of the change in Ca²⁺ activity (from the first to the second imaging session) between groups (i.e. S vs SD vs REMD). Showing a reduction across time in one group but not the others is not the same as comparing groups to each other.

The authors exclude filopodia from their analyses. Spine morphologies fall along a spectrum, making the length/width criteria for distinguishing filopodia from other spines somewhat arbitrary. Please justify the length/width criteria used to distinguish filopodia from spines, and please also explain the biological rationale for excluding filopodia from the analysis.

The authors state that spine analysis was performed blind, but please also clarify whether any non-quantitative portions of the calcium analysis (e.g. manual selection of ROIs around soma and dendrites) were performed blind to sleep/awake state and treatment condition.

The authors state that dendrites were chosen randomly for analysis (Methods, lines 506-507). In our experience, many dendrites are not analyzable due to close proximity to neighboring dendrites, and it would be difficult to randomly select dendrites for analysis. Please state how the random selection was performed (e.g. assignment of numerical IDs to dendrites for random selection of a numerical ID), or please remove this statement from the methods if dendrites were not actually selected randomly.

The authors state that they imaged binocular V1 (Results, lines 98-99), but the authors do not describe any procedures (e.g. intrinsic signal imaging) that would demarcate the boundaries of binocular vs monocular V1. Please clarify how binocular V1 was identified.

In their methods, the authors should state the time of day (relative to the animals' light/dark cycle) when experiments were run, and confirm that time of day did not differ between treatment groups. Sleep deprivation could have different effects depending on the animals' likelihood of sleeping during that time of day under baseline conditions.

The manuscript is so full of abbreviations that it is difficult to read. While the abbreviations make it easier for the writer, they make it harder for the reader. These should be minimized.

We very much appreciate reviewers' constructive suggestions and criticisms. Based on their suggestions, we have modified our manuscript, provided more details on the methodology, and added more data analysis in the revised version. Below are our Point-by-point response (in blue) to reviewers 1-3:

Reviewer #1:

This is a very interesting paper that explores a topic that is important to the field of sleep research specifically and neuronal plasticity more generally. The authors present a compelling case to support their hypotheses. The experiments are solid and the interpretation is, generally, in keeping with the data. It is important to note that the results provide a coherent and consistent picture. I am very enthusiastic about this data. This manuscript fills a hole in the field and is impressive. I do not have any major concerns.

We sincerely thank the reviewer for her/his time in reviewing the manuscript and for the very strong support of our work.

Reviewer #2:

Overall it is very interesting, well done and important study. Here, using two-photon imaging, Y. Zhou et al demonstrate that high levels of Ca activity observed during REM sleep in one-month old mice contribute/lead to elimination of dendritic spines in visual and prefrontal cortices. The results are convincing. Despite my very positive evaluation of the study, I have several suggestions that will improve the presentation of findings.

Major.

1. Both, title and abstract do not contain any mentioning that the study was done during critical period of cortical development. Therefore, it might provide misleading information to a not careful reader.

As the reviewer pointed out correctly, we did not emphasize that the study was done during critical period of cortical development in the original submission. We only mentioned once in the abstract "We found that monocular deprivation (MD) or auditory-cued fear conditioning (FC) caused rapid dendritic spine elimination in the developing V1 or FrA". Because all the mice used in the study were 1-month-old and we were not sure about the exact critical period for FrA development, we did not state "critical period of cortical development" in the title or abstract.

In the revised abstract, we now specifically mention the age of mice ("1-month-old") and also state "the developing V1 or FrA" twice. We hope that these modifications would make it clear that the work was done in the developing cortex.

2. The result section does not contain much of numerical info. In each section of results, please indicate how many of animals, dendrites or spines were used to obtain each piece of data.

As suggested, we have added the number of animals, dendrites or spines in the each section of results (either in the main text or figure legends).

Minor.

1. Videos. Add calibration bar throughout recording (not just on the first frame). Add time indicator. If possible, play it slower that the reader can clearly see activity. It appears that NREM is very brief in both videos (order of 10 sec if the movie was 40 sec). Could you show more convenient segments?

As suggested, we have added calibration bar and time throughout recordings and played them slower. In both videos, quiet awake, NREM sleep and REM sleep have the same duration (~40 seconds for each).

2. Fig. 6 c, e. Please indicate units.

Y axis in Fig. 6c or e represents the integrated dendritic Ca^{2+} activity ($\Delta F/F$) over 1 minute (100%). We now add this information on the Y axis to indicate units.

3. Indicate injected volume of MK801.

We have now added the injected volume of MK801 (~40 nl) in the method section.

4. Bone thinning described twice in the method section.

We now mention bone thinning once in the method section of *Surgery for imaging and EEG/EMG recording* and remove bone thinning in the section of *Imaging dendritic spine remodeling*.

We sincerely thank the reviewer for her/his time in reviewing the manuscript and for all valuable comments and suggestions, which helped us to improve the quality of the manuscript.

Reviewer #3:

General Comments: In this manuscript, the authors use in vivo structural and functional imaging to show that REM sleep deprivation, or blockade of NMDARs during REM sleep, alters

spine remodeling in the context of monocular deprivation and fear conditioning in visual (V1) and prefrontal cortices (FC), respectively. The results complement prior work on the effects of sleep deprivation in various sensory and behavioral paradigms, including prior work from the same group showing that REM sleep deprivation or blockade of NMDARs during REM sleep alters motor learning-induced spine remodeling (Li et al., Nat. Neurosci. 2017). There are several reasons why this study is inappropriate for Nature Communications. First and foremost is the limited novelty. The work is incremental, at best, in relation to previously published work from the same group and provides no further conceptual insight.

We believe that our findings reveal novel and important functions of REM sleep in experience-dependent dendritic spine elimination, which has never been addressed before. *The type of experiences and experience-dependent plasticity, brain regions involved, and the effects of REM sleep are all different between the current study and previously published work (e.g., Li et al., Nat. Neurosci. 2017). None of the findings in the current manuscript could be predicted from previously published study (Li et al., Nat. Neurosci. 2017).* Consequently, this work provides novel conceptual insight into REM functions that was not revealed in previously published work, as detailed below:

(1) The current study investigated the role of REM sleep in fear conditioning or visual deprivation-induced elimination of *existing* spines in FrA and V1. Previous work by Li et al. (Li et al., 2017) studied the role of REM sleep in selective maintenance of new spines induced by motor training in motor cortex. Unlike fear conditioning and visual deprivation, motor training did not increase elimination of existing spines within 8-48 hours (Yang et al., 2014). For this reason, the study by Li et al (Li et al., 2017) did not attempt to investigate the role of REM sleep in elimination of existing spines within the first 8 hours after motor training. Because there is no evidence that REM sleep affects elimination of existing spines after motor training by Li et al (2017), our findings that REM sleep has a role in the elimination of existing spines after fear conditioning or visual deprivation are entirely novel, not just incremental, in relation to previously published work.

(2) The effect of REM sleep on neuronal activity has not been studied in the motor cortex after motor training in Li et al (Li et al., 2017). To our knowledge, our study shows, for the first time, that REM sleep reduces experience-dependent pyramidal neuronal activity (after fear conditioning or visual deprivation) in the developing cortex.

(3) Learning-induced reorganization of neuronal connections could occur by adding new connections, removing existing connections or both. Conceptually, it is not known whether REM sleep affects one or both of these processes. Previous studies reported that motor training increases new spine formation over 8-48 hours without increasing the elimination rate of existing spines. REM sleep facilitates the selective maintenance of new spines that are

formed over 8 hours after motor training (Li et al., 2017). In contrast, fear conditioning and visual deprivation predominately increase elimination of existing spines within 8-48 hours. Our study shows that REM sleep facilitates this removal of existing spines after fear conditioning and visual deprivation. *Together with previous work (Li et al., 2017), the current study indicates that REM sleep facilitates both the addition of new synapses and the removal of existing synapses, dependent on the types of experience that precede sleep.* Such a conceptual insight could not be gained alone from previous studies involving motor learning.

Based on the aforementioned differences between the current study and previously published work, we respectfully disagree with the reviewer's comments " First and foremost is the limited novelty. The work is incremental, at best, in relation to previously published work from the same group and provides no further conceptual insight." We believe that the findings of REM sleep functions in experience-dependent elimination of synaptic connections and reduction of neuronal activity in the cortex are novel and important in the fields of sleep and synaptic plasticity. We think that our view is consistent with that of the other two reviewers.

One of the main claims, repeated in both the abstract, results, and discussion, that blockade of dendritic calcium spikes specifically during REM sleep prevents MD- and FC-induced dendritic spine elimination in V1 and FC, is unsubstantiated (more below). Finally, there are major concerns related to the rigor of experimental procedures and the authors' transparency about their experimental procedures, as well as animal welfare. We outline these and other concerns below.

We have provided our point-by-point responses to the above comments below.

Major concerns:

This study primarily extends the authors' own previously published work from motor cortex (Li et al., Nat. Neurosci. 2017) to two additional cortical regions and behavioral/sensory paradigms. The authors over represent its novelty in their Introduction and Discussion by referring to their previous work in terms of spine maintenance and the current work in terms of spine elimination (Introduction, lines 77-79; Discussion, lines 255-256).

As outlined in the above response to the reviewer's general comments, we believe that our study does reveal the novel role of REM sleep in experience-dependent synapse elimination. To our knowledge, the effect of REM sleep in experience-dependent synapse elimination has not been investigated previously (see also below).

The authors' statement that "the role of sleep in experience-dependent elimination of existing synapses has not been examined previously" (lines 255-256) is untrue, since the authors' own published work shows effects of REM sleep deprivation and MK-801 treatment on spine

elimination in the context of motor learning (Li et al., Nat. Neurosci. 2017). In fact, their 2017 paper distinguishes elimination of pre-existing spines from elimination of new spines, while the current study cannot make such distinctions due to the use of only 2 imaging sessions for most experiments.

We maintain that our statement that “the role of sleep in experience-dependent elimination of existing synapses has not been examined previously” (lines 255-256) is *true*, not *untrue*. Our previously published work (Li et al., 2017) investigated the role of REM sleep in eliminating and strengthening *newly formed spines* induced by motor training. *Because motor training did not increase elimination of existing spines within 8-48 hours (Yang et al., 2014), the role of REM sleep in elimination of existing spines after motor training was not investigated in the study by Li et al. (2017).* The major finding in the study by Li et al (2017) is that REM sleep facilitates selective maintenance of new spines that are formed 8 hours after motor training. In fact, in that study, REM sleep has no significant effect on the elimination of pre-existing spines (existing spines persisted 8 hours after motor training). If anything, this observation would suggest that REM sleep may not have an effect on elimination of existing spines induced by fear conditioning and visual deprivation, *in contrast to our findings.*

In addition, while the previous study by Li et al (2017) found the role of REM sleep in eliminating and strengthening newly-formed spines after motor training, it is important to note that newly-formed spines and existing spines are different in many aspects such as molecular compositions, size and life time. Based on the REM function in eliminating and strengthening new spines that are formed 8 hours after motor training, one should not assume that REM sleep has similar roles in the elimination of existing spines after fear conditioning or visual deprivation.

The MK-801 effects in Fig. 6 suggests that NMDARs mediate the effect of REM sleep on Ca spikes and experience-dependent spine remodeling. However, it is not possible to conclude that this is sequential, where NMDAR blockade eliminates Ca spikes and that results in reduced spine elimination during REM sleep. An alternative interpretation it that the MK801 treatment, by blocking NMDARs independently influences both Ca spikes and spine elimination. Indeed, as stated in the last paragraph of the discussion “Further studies are needed to examine the generation of dendritic Ca²⁺ spikes and down-stream signaling pathways to further delineate the role of REM sleep in modulating synaptic connectivity in development and learning.”

We show in Fig. 5 a substantial increase in dendritic Ca²⁺ spikes in both V1 and FrA during REM sleep as compared to other brain states. Because dendritic Ca²⁺ spikes are important for synaptic potentiation and depotentiation (Golding et al., 2002; Holthoff et al., 2004; Kampa et al., 2006; Nevian and Sakmann, 2004; Sheffield and Dombeck, 2015), we then tested the hypothesis that dendritic Ca²⁺ spikes during REM sleep are involved in dendritic spine

elimination induced by MD and FC by locally injecting MK801 in the FrA or V1 during REM sleep or non-REM sleep (as a control). Based on the findings in Fig. 6 and given the function of dendritic Ca^{2+} spikes in synaptic plasticity, the effect of MK801 injection on spine elimination is likely related to the blockade of dendritic Ca^{2+} spikes during REM. We therefore concluded "these results suggest that dendritic calcium spikes arising during REM sleep play an important role in facilitating experience-dependent dendritic spine elimination."

We agree with the reviewer that MK801 blockade of NMDAR not only reduces dendritic Ca^{2+} spikes but also could influence spine elimination independently of dendritic Ca^{2+} spikes via non-ionicotropic NMDA receptor signalling (Nabavi et al., 2013; Stein et al., 2015). In the latter scenario, because injecting MK801 during REM sleep, not during non-REM sleep, reduces spine elimination, MK801 blockade of NMDAR (independent of dendritic Ca^{2+} spikes) must trigger downstream signalling pathways during REM differently from during NREM, leading to spine elimination only during REM. At the moment, we do not have evidence to support or against this latter scenario or other possibilities. We have now mentioned the potential involvement of non-ionicotropic NMDA receptor signalling (Nabavi et al., 2013; Stein et al., 2015) and tone down the conclusion to "these results suggest that dendritic Ca^{2+} spikes arising during REM sleep are involved in facilitating experience-dependent dendritic spine elimination". We hope the reviewer is fine with these changes.

The experimental design in most of the experiments lack important baseline controls. In the same way that supplementary fig. 3 shows controls for Figs 2&3 - a 4-hour imaging interval with sleep manipulation but without MD, there should be an equivalent control for Figs 3&4 - prefrontal cortex without fear conditioning. Same for Fig 6, a control experiment is warranted testing whether MK-801 during REM sleep reduces spine elimination under baseline conditions, in animals that are not undergoing monocular deprivation or fear conditioning.

The major question we are addressing is whether sleep/REM sleep has a role in fear conditioning/visual deprivation-dependent spine elimination. Regardless of the outcome, the baseline data would not change the conclusions of the work. For example, data in supplementary fig. 3 suggests that REM sleep does not have significant effects on spine elimination or formation over 4 hours in control mice (without MD). If the results turned out differently, we would still make the same conclusion that REM sleep has a role in visual deprivation-dependent spine elimination. We added supplementary fig. 3 in the manuscript because we did this experiment early on to explore the effect of REM sleep in the development of visual cortex. This data is consistent with, but not needed for, the conclusion on REM sleep function in visual deprivation-dependent spine elimination.

The baseline data suggested by the reviewer would involve lots of work (nearly double the amount of work) and the outcome would not change our major conclusions of the

manuscript. In addition, we feel it would be difficult to interpret the baseline controls. If REM or MK801 has an effect on control mice without MD or fear conditioning, we would not know whether we should interpret such results as REM function in spine development or plasticity induced by experience in home cages. For the reasons above, we do not think the baseline controls are necessary to be included in the manuscript.

The authors state that for somatic calcium imaging during sleep and awake states, “the skull overlying the cortex of interest was removed and replaced with a glass window right before two-photon imaging” (Methods, lines 493-494). It is unclear why the authors would perform cranial window surgery immediately before imaging, since cranial windows are routinely imaged for days or weeks after surgery (when using a glass implant, as the authors did for somatic calcium imaging). This is particularly concerning for an experiment involving functional imaging in awake animals, since recent major surgery under anesthesia may affect neuronal activity and responsiveness to the visual and auditory cues presented during imaging. Further, the authors themselves have previously published work showing inflammation immediately post cranial window surgery (Xu et al., 2007) that is only mitigated after a 1-2 weeks recovery (Holtmaat et al., 2009). This major confound could be avoided simply by giving the animals a recovery period after cranial window surgery.

We would like to emphasize that the surgery and imaging procedures for somatic calcium imaging were performed strictly according to the approved animal protocol, which states "A small skull region (~1 mm in diameter) will be thinned or the craniotomy will be performed with a high-speed drill when the animals are under deep anesthesia. For awake animal imaging, two-photon imaging will be performed 24 hours after cranial window creation" (see also below for more details regarding the surgery and imaging). The same procedures were used in previously published work (Cichon and Gan, 2015; Li et al., 2017; Yang et al., 2014). In the method sections of both *Surgery for imaging and EEG/EMG recording* and *imaging dendritic spine remodeling in awake, head-restrained mice*, we referred to these previous publications and also mentioned that the surgery were performed 24 hours before imaging.

In the method section of *Two-photon Ca^{2+} imaging of apical dendrites and somas of layer 5 pyramidal neurons in mice expressing GCaMP6s*, we wrote "the skull overlying the cortex of interest was removed and replaced with a glass window right before two-photon imaging" (Methods, lines 493-494)". We acknowledge that this sentence is confusing as only a small piece of bone (~200 μ m in diameter) was removed 24 hours before imaging. We apologize for this confusion and have now revised the sentence. We appreciate the reviewer's concern that recent major surgery under anesthesia may affect neuronal activity and responsiveness to the visual and auditory cues presented during imaging. We had the same concern as the reviewer. That was why we first performed the surgery (head-post implantation

and skull thinning or craniotomy) in mice under anesthesia. 24 hours after mice recovered from anesthesia, we then imaged neuronal activity of layer 5 pyramidal neurons in awake mice.

We now realize that we described surgery in several places in the method section (Surgery for imaging and EEG/EMG recording, imaging dendritic spine remodeling, two-photon Ca^{2+} imaging of apical dendrites and somas). This was in part due to slightly different procedures used for imaging dendrites and somas. In the revised manuscript, we describe all the surgical procedures only in the section of Surgery for imaging and EEG/EMG recording, as well as added more details in the surgical/imaging procedures and justifications.

We would also like to take this opportunity to clarify why we perform cranial window surgery 24 hours before imaging, rather than using thinned skull window or allowing 1-2 weeks recovery after cranial window surgery. Due to spherical aberration caused by thinned skull, it is difficult to image somas of layer 5 pyramidal neurons located $> 500 \mu\text{m}$ below the skull. We therefore chose to image the activity of these cells via craniotomy window. As the reviewer pointed out correctly, cranial window surgery over a large region (5 mm in diameter) would cause inflammation that is sometimes mitigated after 1-2 weeks recovery. In our hands, most of the mice undergoing such cranial window surgery still could not be imaged after a 1-2 weeks recovery. This resulted in many unsuccessful experiments and use of animals in the past. On the other hand, we were able to image somas of pyramidal neurons in layer 5 over hours in almost every animal if a small cranial window ($\sim 200 \mu\text{m}$) was made and covered by a coverglass 24 hours (rather than 1-2 weeks) before imaging. We have now added the above information in the method.

This design also raises concern about the welfare of animals undergoing prolonged experiments when the somatic calcium imaging immediately follows the window surgery, involving 2 awake head-fixed imaging sessions on the same day, with fear conditioning and 4 hours of sleep deprivation occurring between the imaging sessions. This is a demanding protocol even for mice that are not recovering from surgery. The authors' description of the timeline of surgery/anesthesia is vague, with only brief mention of the cranial window implantation in a section that focuses primarily on the imaging procedures. The timelines for implanting a head post (for head fixation to the microscope) and habituation to awake head fixation are also not described for the experiments shown in Fig. 2 and Fig. 4. This needs clarification and confirmation that this experimental timeline, including the lack of recovery time from cranial window surgery, was reviewed by the relevant institutional committee(s) on animal care. We see no scientific justification for the lack of surgery recovery time and strongly encourage the authors to give their mice appropriate recovery times in the future.

We appreciate the reviewer's concerns with regards to animal welfare. All the experiments involving surgery and imaging were performed strictly according to the approved animal protocols: "For awake animal imaging, two-photon imaging will be performed at least 24 hours after cranial window creation and mice will be physically restrained during the imaging. Before the imaging experiments start, mice will be acclimated to the imaging apparatus after they have recovered from the anesthesia in order to minimize potential stress due to physical restraint". The same procedures were used in previously published work (Cichon and Gan, 2015; Li et al., 2017; Yang et al., 2014).

As the reviewer pointed out correctly, the timeline of surgery/anesthesia was vague, and we only briefly mentioned the cranial window implantation in a section that focuses primarily on the imaging procedures. The timelines for implanting a head post (for head fixation to the microscope) and habituation to awake head fixation were also not described clearly for the experiments shown in Fig. 2 and Fig. 4. We realize that surgical procedures were described in several places in the method section (Surgery for imaging and EEG/EMG recording, imaging dendritic spine remodeling, two-photon Ca²⁺ imaging of apical dendrites and somas). In the revised manuscript, we now describe the surgeries only in the section of Surgery for imaging and EEG/EMG recording, along with the detailed timeline of surgery/anesthesia including implanting a head mount (for head fixation to the microscope), skull thinning, cranial window surgery, and habituation to awake head fixation. Specifically, we added the following information: Head mount/EEG implantation, skull thinning surgery or a small craniotomy window surgery were performed in anesthetized mice according to the previous studies (Yang et al., 2014; Li et al., 2017). Mice were given at least 24 hours to recover from the surgery-related anesthesia before imaging. Prior to imaging, mice with head mounts were habituated for a few times (10 min each time) in the imaging apparatus to minimize potential stress effects of head restraining. For imaging dendritic structure and activity, a thin layer of silicon was applied to cover the thinned window after the surgery. The silicon was removed for imaging, and mice were imaged under awake and head-fixed condition. For imaging somatic activity of layer 5 pyramidal neurons, a small craniotomy window (~200 μ m in diameter) was made and immediately covered with a glass coverslip in anesthetized mice. 24 hours later, somatic calcium activity of layer 5 pyramidal neurons were imaged under awake and head-fixed condition twice, each lasting for ~30 minutes.

The MK-801 experimental timeline shown in Fig. 6f and Fig. 6h suggests that animals were head-fixed (to enable MK-801 application via the Picospritzer) for the entire 4 hrs between imaging sessions, in addition to being head-fixed during the 2 imaging sessions. Please clarify in the Methods whether the animals remained head-fixed for the entire 4-hour period between imaging sessions, and confirm that the duration of head fixation was reviewed by the relevant institutional committee(s) on animal care. We do understand the scientific justification for this

prolonged head fixation, i.e. enabling use of a Picospritzer that would be infeasible in freely moving animals. However, the authors should be transparent in their Methods about the duration of head fixation. We also encourage the authors to consider in their future studies whether providing a lick spout with water and/or calorific liquid would be appropriate during prolonged head fixation.

We have now clarified in the Methods that the animals remained head-fixed for the entire 4-hour period between imaging sessions: For local application of MK801 (200 μ M in artificial cerebrospinal fluid (ACSF), M107, Sigma-Aldrich), mice were head restrained for 4 hours and a glass microelectrode with a 20- μ m outer diameter was inserted through a bone flap into the superficial layer of the cortex (\sim 60 μ m below the pial surface) with an angle of 30° toward and \sim 100 μ m away from the imaging area.

We also confirm that the duration of head fixation was reviewed and approved by the relevant institutional committee on animal care. Regarding the head-fixed duration in imaging and injection, our approved protocol states “Intracerebral injection and topical administration will be performed during anesthetized imaging or awake imaging. Intracerebral injection and topical administration for awake animal will be performed with head mount after the implantation of head mount for awake imaging.” “EEG/EMG Recording will be performed to monitor the animals' brain activity during sleep, awake and anesthesia.” “For EEG/EMG recordings under the microscope, 50% of animals will be imaged during sleep recording. All animals will be placed on a heating pad. Animals not entering sleep after 1 hour of recording, or awake for more than 1 hour (as indicated on the EEG/EMG monitor) will be removed from the microscope and returned to home cage. The maximum duration of the imaging/recording session during sleep is 4 hours.”

We thank the reviewer for the suggestion of providing a lick spout with water and/or calorific liquid during prolonged head fixation. We typically put wetted food pellets near mice under the microscope. Adding a lick spout with calorific liquid is a good idea.

Minor concerns:

The authors state that they used Photoshop to remove “fluorescent structures near and out of the focal plane” in their example images (Methods, lines 517-518). We understand the difficulty of displaying a single dendrite from a 3D image stack in a 2D example image, but the use of Photoshop to remove fluorescent structures does not comply with Nature Communications' current image processing guidelines (<https://www.nature.com/nature-research/editorial-policies/image-integrity>). We request that the authors produce example images that comply with current image processing guidelines.

In the imaging examples in Fig. 1 and 3, we removed surrounding fluorescently labeled dendrites or axons that are not related to the dendrites in display. This procedure did not alter the data or data quantification itself, and has been done in the field in the past decades. We have now provided 3-D image stacks for 2D example images for figure 1 to show eliminated spines were not due to the removal of fluorescent structures as new supplementary videos 1 and 2. We would be glad to provide more examples upon recommendation from the reviewer.

For calcium imaging of dendrites, the authors should clarify how motion artifacts were accounted for, since they did not use a cell fill to visualize dendrites during times of low Ca^{2+} /GCaMP signal, and even slight physiological motion can cause dendrites to move out of focus in Z. The authors' use of StackReg for image registration (Methods, line 549) suggests that there were XY motion artifacts in their images, but StackReg cannot correct Z movement. In the supplemental videos, there are dendrite segments whose GCaMP fluorescence is only visible during Ca^{2+} spikes, and it is not possible to visually determine whether the dendrites are present with low Ca^{2+} or have moved out of focus when they are not visible in the videos. This issue is particularly concerning when calcium spikes are being compared between different sleep/awake states, since motion artifacts may differ between different sleep/awake states.

For calcium imaging of dendrites, we always made sure that the same focal plane was imaged over time by looking at least 3 small and bright fluorescent structures. An example was shown in the videos provided (circled). We found that when animals were in sleep states (NREM sleep and REM sleep) or quiet awake state, the movement along Z-axis was typically less than 1 μm . StackReg for image registration was used for correcting drifts in XY position which could happen over time. Due to spherical aberration along Z-axis and aided by the fluorescent structures, dendritic spikes of the same dendrites on the same focal plane could be imaged across different sleep/awake states, which is supported by the videos we showed as well as by previous studies (Cichon and Gan, 2015; Li et al., 2017). The increase of Ca^{2+} spikes during REM sleep is very robust and readily observable across different states.

The authors describe their dendritic calcium events as dendritic spikes, but back-propagating action potentials (bAPs) are not distinguishable from dendritic spikes in this imaging paradigm. The MK-801 effects in Fig. 6b-e suggest that at least some of the Ca^{2+} events were dendritic spikes dependent on excitatory synaptic input in L1, but the authors should acknowledge in the text of the manuscript that some of the dendritic Ca^{2+} events they observe could originate from bAPs rather than dendritic spikes.

Dendritic Ca^{2+} events we imaged occurred on apical tuft dendrites of layer 5 pyramidal neurons. Several lines of evidence indicate that these Ca^{2+} spikes could be generated in the absence of back-propagating action potentials (bAPs) (Cichon and Gan, 2015; Larkum and Zhu, 2002). While bAPs originated from somas may not reach apical tuft dendrites (Helmchen et al.,

1999; Larkum et al., 1999), they could contribute to the generation of dendritic Ca^{2+} spikes. We have discussed such a possibility as suggested.

For somatic calcium activity (Fig. 2 and Fig. 4), please clarify in the legend the measure used on the Y-axis (e.g. is this the number of events, or the percent of CS presentations that produced an event, etc).

As suggested by the reviewer, we have clarified in the Figures and legends that the Y-axis was measured by integrated somatic Ca^{2+} activity of layer 5 pyramidal neurons during 60-s visual stimulation in Fig. 2 and 30-s CS presentation in Fig. 4, respectively. The integrated activity was measured as the integrated $\Delta F/F_0$ above threshold over the period of 1 minute or 30 seconds.

For the data shown in Fig. 2 and Fig. 4, the authors should provide a statistical comparison of the change in Ca^{2+} activity (from the first to the second imaging session) between groups (i.e. S vs SD vs REMD). Showing a reduction across time in one group but not the others is not the same as comparing groups to each other.

As suggested, we have provided a statistical comparison of the change in Ca^{2+} activity (from the first to the second imaging session) between groups (i.e. S vs SD vs REMD) in the revised manuscript in the new Supplementary Figs 4 and 6.

The authors exclude filopodia from their analyses. Spine morphologies fall along a spectrum, making the length/width criteria for distinguishing filopodia from other spines somewhat arbitrary. Please justify the length/width criteria used to distinguish filopodia from spines, and please also explain the biological rationale for excluding filopodia from the analysis.

We have now mentioned the criteria used to distinguish filopodia from spines in the Method, as follows: For each dendritic segment analysed, filopodia were identified typically as long, thin protrusions with ratio of head diameter to neck diameter $<1.2:1$ and ratio of length to neck diameter $>3:1$. Short, thin protrusions with ratio of head diameter to adjacent dendritic shaft diameter $<1:2$ and ratio of head intensity to adjacent dendritic shaft intensity $<1:3$ were also identified as filopodia. The remaining protrusions were classified as spines. Such criteria are based on previous published work (Grutzendler et al., 2002; Yang et al., 2014).

We distinguish spines versus filopodia because filopodia are highly dynamic/transient structures and previous studies show monocular deprivation or fear conditioning over days did not cause changes in filopodial dynamics (Lai et al., 2012; Zhou et al., 2017). We therefore did not investigate the effect of sleep/REM sleep on filopodia.

The authors state that spine analysis was performed blind, but please also clarify whether any non-quantitative portions of the calcium analysis (e.g. manual selection of ROIs around soma and dendrites) were performed blind to sleep/awake state and treatment condition.

We have now stated in the Method that for dendritic and somatic calcium activity analysis, we quantified calcium activity of dendrites (>30 μm in length) and somas in images and videos blind to various brain states and treatment conditions.

The authors state that dendrites were chosen randomly for analysis (Methods, lines 506-507). In our experience, many dendrites are not analyzable due to close proximity to neighboring dendrites, and it would be difficult to randomly select dendrites for analysis. Please state how the random selection was performed (e.g. assignment of numerical IDs to dendrites for random selection of a numerical ID), or please remove this statement from the methods if dendrites were not actually selected randomly.

We used the word "randomly" in the sense that we selected dendrites for analysis as long as these dendrites exhibited good signal to noise ratio and did not overlap with dendrites nearby. We have mentioned this selection criterion in the method to replace the statement "dendrites were chosen randomly for analysis".

The authors state that they imaged binocular V1 (Results, lines 98-99), but the authors do not describe any procedures (e.g. intrinsic signal imaging) that would demarcate the boundaries of binocular vs monocular V1. Please clarify how binocular V1 was identified.

The method used to identify the binocular region was described in our recently-published study (Zhou et al., 2017). The binocular region of the primary visual cortex was identified based on stereotaxic coordinates (3.0 mm posterior from bregma and 3.0 mm lateral from the midline). We have now added this information in the method.

In their methods, the authors should state the time of day (relative to the animals' light/dark cycle) when experiments were run, and confirm that time of day did not differ between treatment groups. Sleep deprivation could have different effects depending on the animals' likelihood of sleeping during that time of day under baseline conditions.

We have stated the time of day when experiments were run as follows: 1. In Method, Mice were maintained at 22 ± 2 °C with a 12-h light: dark cycle (lights on at 6:30 am, lights off at 6:30 pm). All experiments were conducted during the light cycle, starting at 8:00 am. 2. In Figures, we have shown the schematic of experimental design of the time when experiments were started.

The manuscript is so full of abbreviations that it is difficult to read. While the abbreviations make it easier for the writer, they make it harder for the reader. These should be minimized.

As suggested by the reviewer, we have minimized the abbreviations throughout the manuscript.

Finally, we sincerely thank the reviewer for his/her constructive criticisms and valuable comments, which are very helpful for us to improve the manuscript. We hope that we have addressed the reviewers' concerns satisfactorily.

References

- Cichon, J., and Gan, W.B. (2015). Branch-specific dendritic Ca²⁺ spikes cause persistent synaptic plasticity. *Nature* 520, 180-185.
- Golding, N.L., Staff, N.P., and Spruston, N. (2002). Dendritic spikes as a mechanism for cooperative long-term potentiation. *Nature* 418, 326-331.
- Grutzendler, J., Kasthuri, N., and Gan, W.B. (2002). Long-term dendritic spine stability in the adult cortex. *Nature* 420, 812-816.
- Helmchen, F., Svoboda, K., Denk, W., and Tank, D.W. (1999). In vivo dendritic calcium dynamics in deep-layer cortical pyramidal neurons. *Nat Neurosci* 2, 989-996.
- Holthoff, K., Kovalchuk, Y., Yuste, R., and Konnerth, A. (2004). Single-shock LTD by local dendritic spikes in pyramidal neurons of mouse visual cortex. *J Physiol* 560, 27-36.
- Holtmaat, A., Bonhoeffer, T., Chow, D.K., Chuckowree, J., De Paola, V., Hofer, S.B., Hubener, M., Keck, T., Knott, G., Lee, W.C., *et al.* (2009). Long-term, high-resolution imaging in the mouse neocortex through a chronic cranial window. *Nat Protoc* 4, 1128-1144.
- Kampa, B.M., Letzkus, J.J., and Stuart, G.J. (2006). Requirement of dendritic calcium spikes for induction of spike-timing-dependent synaptic plasticity. *J Physiol* 574, 283-290.
- Lai, C.S., Franke, T.F., and Gan, W.B. (2012). Opposite effects of fear conditioning and extinction on dendritic spine remodelling. *Nature* 483, 87-91.
- Larkum, M.E., Kaiser, K.M., and Sakmann, B. (1999). Calcium electrogenesis in distal apical dendrites of layer 5 pyramidal cells at a critical frequency of back-propagating action potentials. *Proc Natl Acad Sci U S A* 96, 14600-14604.
- Larkum, M.E., and Zhu, J.J. (2002). Signaling of layer 1 and whisker-evoked Ca²⁺ and Na⁺ action potentials in distal and terminal dendrites of rat neocortical pyramidal neurons in vitro and in vivo. *J Neurosci* 22, 6991-7005.
- Li, W., Ma, L., Yang, G., and Gan, W.B. (2017). REM sleep selectively prunes and maintains new synapses in development and learning. *Nat Neurosci* 20, 427-437.
- Nabavi, S., Kessels, H.W., Alfonso, S., Aow, J., Fox, R., and Malinow, R. (2013). Metabotropic NMDA receptor function is required for NMDA receptor-dependent long-term depression. *Proc Natl Acad Sci U S A* 110, 4027-4032.
- Nevian, T., and Sakmann, B. (2004). Single spine Ca²⁺ signals evoked by coincident EPSPs and backpropagating action potentials in spiny stellate cells of layer 4 in the juvenile rat somatosensory barrel cortex. *J Neurosci* 24, 1689-1699.

Sheffield, M.E., and Dombeck, D.A. (2015). Calcium transient prevalence across the dendritic arbour predicts place field properties. *Nature* *517*, 200-204.

Stein, I.S., Gray, J.A., and Zito, K. (2015). Non-Ionotropic NMDA Receptor Signaling Drives Activity-Induced Dendritic Spine Shrinkage. *J Neurosci* *35*, 12303-12308.

Xu, H.T., Pan, F., Yang, G., and Gan, W.B. (2007). Choice of cranial window type for in vivo imaging affects dendritic spine turnover in the cortex. *Nat Neurosci* *10*, 549-551.

Yang, G., Lai, C.S., Cichon, J., Ma, L., Li, W., and Gan, W.B. (2014). Sleep promotes branch-specific formation of dendritic spines after learning. *Science* *344*, 1173-1178.

Zhou, Y., Lai, B., and Gan, W.B. (2017). Monocular deprivation induces dendritic spine elimination in the developing mouse visual cortex. *Scientific reports* *7*, 4977.

REVIEWERS' COMMENTS:

Reviewer #1 (Remarks to the Author):

I have no further comments

Reviewer #2 (Remarks to the Author):

Excellent study. The authors satisfactory responded to most of my comments. Ne still remaining. I am still confused with videos. The manuscript mentions 4 videos and there is a legend for 4 videos. However, in the journal web site, there are 6 videos, which are not numbered. The last two videos likely correspond to video 3 and video 4 in the manuscript. It is unclear to me which of the first 4 videos correspond to the mentioned video 1 and 2?

I have no other comments.
Igor Timofeev

Reviewer #3 (Remarks to the Author):

While the authors have made substantial improvements to the manuscript, they unfortunately have failed to address several key concerns.

Our primary concern in the first submission was related to misrepresentation of novelty, and this continues to be a concern. We still believe the novelty of the study is not at the level of Nature Communications. It is not clear that there is anything the authors can do to improve on this within the scope of the data presented. We find the authors rebuttal on this point as unpersuasive as the presentation in the manuscript.

We had other concerns about experimental procedures and controls, some of which have been addressed. Other points are addressed in the rebuttal, but not in the manuscript, and some points remain unaddressed.

1) In terms of novelty, in several places in the manuscript and Response to Reviewers, the authors still misrepresent the distinction between the present study and Li et al. (2017). To differentiate from Li et al. (2017), the authors still describe the present study in terms of spine elimination and Li et al. (2017) in terms of spine maintenance. Elimination and maintenance are different ways of describing the same phenomenon, i.e. a spine is maintained as long as it is not eliminated. (The authors themselves used both "elimination" and "maintenance" to describe their own results in Li et al., 2017.) It is true that Li et al. (2017) shows effects on longer-term maintenance, while the current study focuses only on short-term maintenance/elimination. However, the short-term effects of REM sleep deprivation in Li et al. (2017) match the current study, i.e. REM sleep facilitates spine elimination in both studies. The authors claim that their current results are opposite to Li et al. (2017) when they state in their Response to Reviewers, "REM sleep facilitates the selective maintenance of new spines that are formed over 8 hours after motor training (Li et al., 2017). In contrast, fear conditioning and visual deprivation predominately increase elimination of existing spines within 8-48 hours." The increase in maintenance observed in Li et al. was only in the long term (4 days); in the short term (16 hrs or 24 hrs), Li et al. found that REM sleep facilitates spine elimination, matching the present study (Fig. 1 and Fig. 4 of Li et al., 2017). The present study therefore corroborates the findings of Li et al. (2017) in different brain regions, rather than showing anything conceptually new or surprising.

In their Response to Reviewers, the authors also differentiate their results from those of Li et al.

(2017) in terms of elimination of new vs existing spines, stating that the present study focuses on existing spines and Li et al. (2017) focuses on new spines. However, the current study cannot distinguish elimination of new spines from elimination of existing spines due to the use of only 2 timepoints in the analyses. When only 2 timepoints are used, all spines are by definition "existing" when they are present in the first timepoint. The present study does in some cases include more than 2 imaging timepoints, but the analyses shown are always between 2 timepoints (e.g. in Supplemental Fig. 5, the middle timepoint is simply skipped for the 24 hr analysis). So in fact all the data shown is for "existing" spines. If the authors would like to make a distinction between new vs existing spines to differentiate the present study from Li et al. (2017), then they will need to analyze more than 2 timepoints.

Perhaps our disagreement here is also related to the definition of novelty. Despite the obfuscation in the rebuttal related to terminology, the only thing this study really adds as compared to Li et al. (2017) is new brain regions and different behavioral/sensory paradigms, with results that are completely in line with previous findings. I would not consider this novel given the limited conceptual advance.

2) We raised the possibility that MK-801 independently affects calcium spikes and spine elimination, rather than calcium spikes serving as an intermediate step between MK-801 and spine elimination. As the authors admit in their rebuttal, " We agree with the reviewer that MK801 blockade of NMDAR not only reduces dendritic Ca²⁺ spikes but also could influence spine elimination independently of dendritic Ca²⁺ spikes via non-ionotropic NMDA receptor signaling" they go on to outline the two scenarios, then say "At the moment, we do not have evidence to support or against this latter scenario or other possibilities". Indeed, the authors have added a brief statement to this effect in the discussion. However, they still pick their favored scenario, and claim in multiple places that their results show a causal role for calcium spikes as an intermediate step between MK-801 and spine elimination (lines 39-40, lines 83-84, line 239, lines 247-248, lines 253-255, lines 265-266). The wording should be adjusted throughout the manuscript to clarify that what the authors have shown is that MK-801 reduces calcium spikes and also reduces spine elimination, i.e. they have not directly shown that calcium spikes are an intermediate step between MK-801 and spine elimination.

3) We requested additional baseline controls to show that the effects of REM sleep deprivation on spine remodeling in FrA are experience-dependent. The authors state in their Response to Reviewers that even if REM sleep deprivation does reduce spine elimination in FrA under baseline conditions, in the absence of fear conditioning, their conclusions still stand. It is customary to use the term "experience-dependent" to refer to phenomena that do not occur in the absence of the relevant experiences. If the authors are not going to provide controls to show that the effects of REMd in FrA are experience-dependent, then they should rephrase their claims and include a discussion of the possibility that sleep affects these measures in a more generalized manner that is not experience-dependent.

4) One of our major concerns in the first submission was related to animal welfare and residual effects of anesthesia during awake imaging, since the first submission stated that surgery was performed "right before two-photon imaging." We thank the authors for the new methods details clarifying the timelines of surgery and imaging, including the 24 hr recovery time between surgery and imaging. The authors state that they used 24 hrs rather than 1-2 weeks recovery because many of their windows cannot be imaged more than 24 hours after surgery due to compromised window clarity. Many labs routinely image weeks or even months after surgery, including windows placed over V1, and the authors' issues with window clarity suggest issues with their surgical technique (e.g. infection or excessive damage to the dura). However, since both groups of mice underwent the same surgical procedure and recovery time, we do not believe that poor surgery

quality explains any group differences in the present study, but we do recommend exploring ways to improve craniotomy surgery quality for future studies.

5) Regarding the use of Photoshop to remove other fluorescent structures from example images: we recognize the difficulty of finding isolated example branches and representing 3D information in a 2D example image. In general, we think it is informative for readers to see images that represent how crowded or isolated the dendrites are in a realistic way, so that readers can assess the relative ease of distinguishing spines from passing neurites in the images used for analysis. The authors' inclusion of unedited 3D images stacks in the new supplementary material is helpful in this respect. In terms of journal-specific image processing standards for the example images shown in Fig 1a, we leave this decision to the editor's discretion.

Point-by-point response to reviewers 2 and 3:

(Our responses in blue)

Reviewer #2:

Excellent study. The authors satisfactory responded to most of my comments. Ne still remaining.

I am still confused with videos. The manuscript mentions 4 videos and there is a legend for 4 videos. However, in the journal web site, there are 6 videos, which are not numbered. The last two videos likely correspond to video 3 and video 4 in the manuscript. It is unclear to me which of the first 4 videos correspond to the mentioned video 1 and 2?

We apologize for the confusion. The first four videos are newly-added ones showing 3-D image stacks of dendrites (in response to the reviewer 3's previous comment). The last two videos show Ca²⁺ imaging of apical tuft dendrites of layer 5 pyramidal neurons expressing the genetically encoded Ca²⁺ indicator GCaMP6s in V1 and FrA under various brain states. We now make sure that the 6 videos are numbered in the Journal web site, and that there is a legend for each video.

Reviewer #3 (Remarks to the Author):

While the authors have made substantial improvements to the manuscript, they unfortunately have failed to address several key concerns. Our primary concern in the first submission was related to misrepresentation of novelty, and this continues to be a concern. We still believe the novelty of the study is not at the level of Nature Communications. It is not clear that there is anything the authors can do to improve on this within the scope of the data presented. We find the authors rebuttal on this point as unpersuasive as the presentation in the manuscript.

We respectfully disagree with the reviewer 3's primary concern about the novelty of our work. We also believe that both reviewers 1 and 2 are in agreement with us. Both of them commented that our study was novel and excellent.

1) In terms of novelty, in several places in the manuscript and Response to Reviewers, the authors still misrepresent the distinction between the present study and Li et al. (2017). To differentiate from Li et al. (2017), the authors still describe the present study in terms of spine elimination and Li et al. (2017) in terms of spine maintenance. Elimination and maintenance are different ways of describing the same phenomenon, i.e. a spine is maintained as long as it is not eliminated. (The authors themselves used both "elimination" and "maintenance" to describe their own results in Li et al., 2017.) It is true that Li et al. (2017) shows effects on longer-term maintenance, while the current study focuses only on short-term maintenance/elimination. However, the short-term effects of REM sleep deprivation in Li et al. (2017) match the current study, i.e. REM sleep facilitates spine elimination in both studies. The authors claim that their current results are opposite to Li et al. (2017) when they state in their Response to Reviewers, "REM sleep facilitates the selective maintenance of new spines that are formed over 8 hours after motor training (Li et al., 2017). In contrast, fear conditioning and visual deprivation predominately increase elimination of existing spines within 8-48 hours." The increase in maintenance observed in Li et al. was only in the long term (4 days); in the short term (16 hrs or 24 hrs), Li et al. found that REM sleep facilitates spine elimination, matching the present study (Fig. 1 and Fig. 4 of Li et al., 2017). The present study therefore corroborates the findings of Li et al. (2017) in different brain regions, rather than showing anything conceptually new or surprising.

In the study by Li et al. (2017), new spines are formed in response to motor training and subsequent REM sleep selectively eliminates and strengthened these new spines. The main finding in Li et al. (2017) is that REM sleep helps the survival of new spines induced by motor training. In our current study, existing spines are formed before new experience (i.e. not related to fear conditioning or visual deprivation). Fear learning or visual deprivation causes the elimination of existing spines in the process that is dependent on REM sleep. The main point in our present study is that REM sleep helps the elimination of existing spines after fear learning and visual deprivation. It is not possible to predict the results in our current study from the previous one (Li et al., 2017). Conceptually, learning-induced reorganization of neuronal connections could occur by adding new connections (in the case of motor training) or removing existing connections (in the case of fear conditioning or visual deprivation). Together with previous work (Li et al., 2017), the current study indicates that REM sleep facilitates changes of synaptic connections (either adding new stable connections or removing existing ones) that is dependent on the types of experience that precede sleep. Such conceptual insight could not be gained alone from previous studies involving motor learning.

It is true that the short-term effects of REM sleep deprivation in Li et al. (2017) include new spine elimination and strengthening. But these new spines are specifically induced by motor training. On the other hand, in our present study, REM sleep facilitates the elimination of spines that exist before mice are subjected to new experience (fear conditioning or visual deprivation). Given many differences between the two studies (motor versus visual/FrA, motor training versus visual deprivation/fear conditioning, new spines versus existing spines, spine formation versus elimination.), we respectfully disagree with the reviewer statement “the present study therefore corroborates the findings of Li et al. (2017) in different brain regions, rather than showing anything conceptually new or surprising”.

In their Response to Reviewers, the authors also differentiate their results from those of Li et al. (2017) in terms of elimination of new vs existing spines, stating that the present study focuses on existing spines and Li et al. (2017) focuses on new spines. However, the current study cannot distinguish elimination of new spines from elimination of existing spines due to the use of only 2 timepoints in the analyses. When only 2 timepoints are used, all spines are by definition “existing” when they are present in the first timepoint. The present study does in some cases include more than 2 imaging timepoints, but the analyses shown are always between 2 timepoints (e.g. in Supplemental Fig. 5, the middle timepoint is simply skipped for the 24 hr analysis). So in fact all the data shown is for “existing” spines. If the authors would like to make a distinction between new vs existing spines to differentiate the present study from Li et al. (2017), then they will need to analyze more than 2 timepoints.

In Li et al. (2017), new spines were specifically induced by motor training between 0 and 8 hours (2 imaging time points). The effect of REM sleep on these newly-formed spines was subsequently examined at 16 or 24 hours (third imaging time point). The current study addresses a completely different question – does REM sleep has a role in fear conditioning/visual deprivation-induced elimination of existing spines? Existing spines are defined here as any spines existed before mice were subjected to fear conditioning or visual deprivation. If I understand the reviewer correctly, she or he wanted us to address a new question --- does REM sleep has a role in fear conditioning/visual deprivation-induced elimination of NEW spines? If so, what is the definition of new spines here? Are they new spines formed a few hours or a few days before fear conditioning or visual deprivation? This seems to be a new question that needs to be better defined, and the significance of addressing such a question is somewhat obscure.

Perhaps our disagreement here is also related to the definition of novelty. Despite the obfuscation in the rebuttal related to terminology, the only thing this study really adds as compared to Li et al. (2017) is new brain regions and different behavioral/sensory paradigms, with results that are completely in line with previous findings. I would not consider this novel given the limited conceptual advance.

As mentioned above, we respectfully disagree with the reviewer's assessment on the novelty of our work. The findings of REM sleep functions in experience-dependent elimination of synaptic connections and reduction of neuronal activity in the cortex are novel and important in the fields of sleep and synaptic plasticity. We think that this view is also supported by the other two reviewers.

2) We raised the possibility that MK-801 independently affects calcium spikes and spine elimination, rather than calcium spikes serving as an intermediate step between MK-801 and spine elimination. As the authors admit in their rebuttal, "We agree with the reviewer that MK801 blockade of NMDAR not only reduces dendritic Ca²⁺ spikes but also could influence spine elimination independently of dendritic Ca²⁺ spikes via non-ionotropic NMDA receptor signaling" they go on to outline the two scenarios, then say "At the moment, we do not have evidence to support or against this latter scenario or other possibilities". Indeed, the authors have added a brief statement to this effect in the discussion. However, they still pick their favored scenario, and claim in multiple places that their results show a causal role for calcium spikes as an intermediate step between MK-801 and spine elimination (lines 39-40, lines 83-84, line 239, lines 247-248, lines 253-255, lines 265-266). The wording should be adjusted throughout the manuscript to clarify that what the authors have shown is that MK-801 reduces calcium spikes and also reduces spine elimination, i.e. they have not directly shown that calcium spikes are an intermediate step between MK-801 and spine elimination.

We state in the discussion that "Because REM sleep duration is short (~1-2 minutes in mice) and spine elimination does not seem to occur during the brief REM sleep episode (unpublished observations), it is likely that dendritic Ca²⁺ spikes during REM sleep set in motion downstream Ca²⁺ signalling cascades that could facilitate the elimination of existing spines triggered by experiences. Furthermore, it is important to note that MK801 blockade of NMDA receptors could influence spine elimination not only by reducing dendritic Ca²⁺ spikes but also via non-ionotropic NMDA receptor signaling (refs 59,60). Further studies are needed to examine the generation of dendritic Ca²⁺ spikes and downstream signalling pathways to further delineate the role of REM sleep in modulating synaptic connectivity in development and learning". We hope these discussion points make it clear to readers that we have not directly shown that calcium spikes are an intermediate step between MK-801 and spine elimination.

3) We requested additional baseline controls to show that the effects of REM sleep deprivation on spine remodeling in FrA are experience-dependent. The authors state in their Response to Reviewers that even if REM sleep deprivation does reduce spine elimination in FrA under baseline conditions, in the absence of fear conditioning, their conclusions still stand. It is customary to use the term "experience-dependent" to refer to phenomena that do not occur in the absence of the relevant experiences. If the authors are not going to provide controls to show that the effects of REMd in FrA are experience-dependent, then they should rephrase their claims and include a discussion of the possibility that sleep affects these measures in a more generalized manner that is not experience-dependent.

Our data support the conclusion -- REM sleep promotes fear or visual experience-dependent dendritic spine elimination in the mouse cortex. We therefore discussed these findings in the context of

experience-dependent synaptic plasticity. Because we do not have data to show whether or not REM sleep promotes experience-independent dendritic spine elimination in the mouse cortex, we would prefer to focus our discussion on what we know. We could add a sentence in the discussion such as “REM sleep may also affect spine elimination in a more generalized manner that is not experience-dependent”. We would leave this for the editors to decide.

4) One of our major concerns in the first submission was related to animal welfare and residual effects of anesthesia during awake imaging, since the first submission stated that surgery was performed “right before two-photon imaging.” We thank the authors for the new methods details clarifying the timelines of surgery and imaging, including the 24 hr recovery time between surgery and imaging. The authors state that they used 24 hrs rather than 1-2 weeks recovery because many of their windows cannot be imaged more than 24 hours after surgery due to compromised window clarity. Many labs routinely image weeks or even months after surgery, including windows placed over V1, and the authors’ issues with window clarity suggest issues with their surgical technique (e.g. infection or excessive damage to the dura). However, since both groups of mice underwent the same surgical procedure and recovery time, we do not believe that poor surgery quality explains any group differences in the present study, but we do recommend exploring ways to improve craniotomy surgery quality for future studies.

We apologize for the potential confusion here regarding our surgical technique for imaging. We did not have any problem of imaging spines when the first imaging session started weeks after the open-skull surgery (although the successful rate varies among labs). However, we (as well as many others) did have problem of imaging spines 2 days after the open-skull surgery. Therefore, we chose to use 24 hours rather than 1-2 weeks recovery time after surgery in our study.

5) Regarding the use of Photoshop to remove other fluorescent structures from example images: we recognize the difficulty of finding isolated example branches and representing 3D information in a 2D example image. In general, we think it is informative for readers to see images that represent how crowded or isolated the dendrites are in a realistic way, so that readers can assess the relative ease of distinguishing spines from passing neurites in the images used for analysis. The authors’ inclusion of unedited 3D images stacks in the new supplementary material is helpful in this respect. In terms of journal-specific image processing standards for the example images shown in Fig 1a, we leave this decision to the editor’s discretion.

We have now followed the Journal’s instructions for presenting imaging data, as outlined in the reporting summary and editorial policy checklist.